# CONTEXTUAL INVERSE REINFORCEMENT LEARNING

## ABSTRACT

We consider the Inverse Reinforcement Learning problem in Contextual Markov Decision Processes. In this setting, the reward, which is unknown to the agent, is a function of a static parameter referred to as the context. There is also an "expert" who knows this mapping and acts according to the optimal policy for each context. The goal of the agent is to learn the expert's mapping by observing demonstrations.

We define an optimization problem for finding this mapping and show that when it is linear, the problem is convex. We present and analyze the sample complexity of three algorithms for solving this problem: the mirrored descent algorithm, evolution strategies, and the ellipsoid method. We also extend the first two methods to work with general reward functions, e.g., deep neural networks, but without the theoretical guarantees. Finally, we compare the different techniques empirically in driving simulation and a medical treatment regime.

## 1 INTRODUCTION

We study sequential decision-making in a Contextual Markov Decision Process (CMDP, Hallak et al. (2015)), where the reward, while unknown to the agent, depends on a static parameter referred to as the *context*. For a concrete example, consider the dynamic treatment regime (Chakraborty & Murphy, 2014). Here, there is a patient and a clinician which acts to improve the patient's health. The context is composed of static information of the patient (such as age and weight); the state is composed of the patient's dynamic measurements (such as heart rate and blood pressure); and the clinician's actions are a set of intervention categories (e.g., infusion). The reward is different for each patient (context), and there is a mapping from the context to the reward.

Recent trends in personalized medicine motivate this model – instead of treating the "average patient", patients are separated into different groups for which the medical decisions are tailored (Fig. 1b). For example, in Wesselink et al. (2018), the authors study organ injury, which may occur when a specific measurement (mean arterial pressure) decreases below a certain threshold. They found that this threshold varies across different patient groups (context). In other examples, clinicians set treatment goals for the patients, i.e., they take actions to make the patient measurements reach some pre-determined values. For instance, in acute respiratory distress syndrome (ARDS), clinicians argue that these treatment goals should depend on the static patient information (the context) (Berngard et al., 2016).

There are serious issues when trying to manually define a reward signal in real-world tasks. When treating patients with sepsis, for example, the only available signal is the mortality of the patient at the end of the treatment (Komorowski et al., 2018). This signal is sparse, and it is unclear how to manually tweak the reward to maximize the patient's health condition (Leike et al., 2017; Raghu et al., 2017; Lee et al., 2019).

To address these issues, we propose the **Contextual Inverse Reinforcement Learning (COIRL)** framework. Similarly to Inverse Reinforcement Learning (Ng & Russell, 2000, IRL), we focus on trying to infer the mapping from contexts to rewards by observing experts. The main challenge in our problem is that for each context there is a different reward, hence, a different optimal policy for each context. Therefore, Apprenticeship Learning algorithms (Abbeel & Ng, 2004; Syed & Schapire, 2008) that try to mimic the expert cannot be used and, instead, we focus on directly learning the mapping.

In particular, our **main contributions** are:

1. We formulate COIRL with a linear mapping as a convex optimization problem.
2. We propose and analyze the sample complexity of three algorithms for COIRL: the mirrored descent alg. (MDA), evolution strategies (ES), and the ellipsoid method.
3. For nonlinear mappings, we implement a deep learning version for MDA and ES (without theoretical guarantees).
4. We compare these methods empirically on two frameworks: an autonomous driving simulator (Abbeel & Ng, 2004) and a dynamic treatment regime (Komorowski et al., 2018).

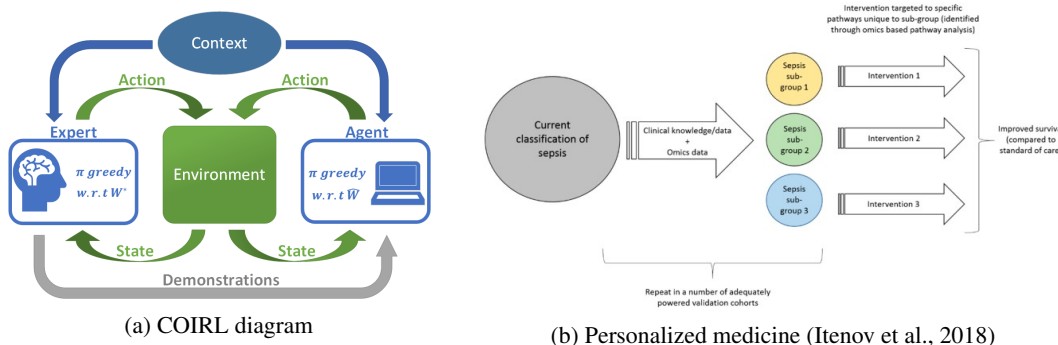

(a) COIRL diagram

(b) Personalized medicine (Itenov et al., 2018)

Figure 1: The COIRL framework (left): a context vector parametrizes the environment. For each context, the expert uses the true mapping from contexts to rewards, $W^*$, and provides demonstrations. The agent learns an estimation of this mapping $\hat{W}$ and acts optimally with respect to it.

## 2 PRELIMINARIES

**Contextual MDPs:** A Markov Decision Process (Puterman, 1994, **MDP**) is defined by the tuple $(\mathcal{S}, \mathcal{A}, P, \xi, R, \gamma)$ where $\mathcal{S}$ is a finite state space, $\mathcal{A}$ a finite action space, $P : S \times S \times A \to [0, 1]$ the transition kernel, $\xi$ the initial state distribution, $R : \mathcal{S} \to \mathbb{R}$ the reward function and $\gamma \in [0, 1)$ is the discount factor. A Contextual MDP (Hallak et al., 2015, **CMDP**) is an extension of an MDP, and is defined by $(\mathcal{C}, \mathcal{S}, \mathcal{A}, \mathcal{M}, \gamma)$ where $\mathcal{C}$ is the context space, and $\mathcal{M}$ is a mapping from contexts $c \in \mathcal{C}$ to MDPs: $\mathcal{M}(c) = (\mathcal{S}, \mathcal{A}, P, R^c, \xi, \gamma)$. In addition, each state is associated with a feature vector $\phi : \mathcal{S} \to [0, 1]^k$. Note that $P$ and $\xi$ are not context dependent.

We consider a setting in which the reward for context $c$ is a linear combination of the state features: $R_c^*(s) = f^*(c)^T \phi(s)$. The goal is to approximate $f^*(c)$ using a function $f_W(c)$, with parameters $W$. This notation allows us to present our algorithms for any function approximator $f_W(c)$, and in particular, a deep neural network (DNN). For the theoretical analysis, we will further assume a *linear setting*, where $f^*(c) = c^T W^*$, $f_W(c) = c^T W$ and that $W^*$ is in some convex set $\mathcal{W}$.

We assume that $c \in \mathcal{C} = \Delta_{d-1}$, the standard $d - 1$ dimensional simplex. This assumption makes the contexts bounded (which we use in our proofs), and it also allows a straight-forward expansion to a model in which the transitions are also a linear mapping of the context (Modi et al., 2018). One way of viewing this model is that each row in the mapping $W^*$ is a base rewards coefficient vector, and the reward for a specific context is a convex combination of these base rewards.

We consider deterministic policies $\pi : \mathcal{S} \to \mathcal{A}$ which dictate the agent's behaviour at each state. The value of a policy $\pi$ for reward coefficients vector $r$ is: $V_r^\pi = E_{\xi,P,\pi}[\sum_{t=0}^{\infty} \gamma^t R(s_t)] = r^T \mu(\pi)$ where $\mu(\pi) := E_{\xi,P,\pi}[\sum_{t=0}^{\infty} \gamma^t \phi(s_t)] \in \mathbb{R}^k$ is called the *feature expectations* of $\pi$. For the optimal policy with respect to (w.r.t.) a reward coefficients vector $r$, we denote the value by $V_r^*$. For any context $c$, $\pi_c^*$ denotes the optimal policy w.r.t. reward $R_c^*(s) = f^*(c)^T \phi(s)$ and $\hat{\pi}_c(W)$ denotes the optimal policy w.r.t. reward $\hat{R}_c(s) = f_W(c)^T \phi(s)$.

**Inverse Reinforcement Learning in CMDPs:** In standard IRL, the goal is to learn a reward which best explains the behavior of an observed expert. The model describing this scenario is the MDP\R -

an MDP without a reward function (also commonly called a controlled Markov chain). Similarly, we denote a CMDP without a mapping of context to reward by **CMDP\M**. The goal in Contextual IRL is to approximate the mapping $f^*(c)$ by observing an expert. The expert knows $f^*(c)$, and for each context $c$, can provide a demonstration from $\pi_c^*$.

**Contextual dynamics:** Learning a transition kernel and an initial state distribution that is parametrized by the context is an orthogonal problem to COIRL. Therefore, we focus only on a contextual reward which simplifies our analysis. Existing methods, such as in Modi et al. (2018), can be used to learn the mappings for the transition kernel and initial distribution in a contextual model. In conjunction with the simulation lemma (Kearns & Singh, 2002), these methods can extend our results to the more general CMDP setting.

## 3  OPTIMIZATION METHODS FOR COIRL

In this section, we propose and analyze optimization algorithms for minimizing the following loss function; Lemma 1 below justifies its use for COIRL.

$$\text{Loss}(W) = \mathbb{E}_c \max_\pi \left[ f_W(c) \cdot \left( \mu(\pi) - \mu(\pi_c^*) \right) \right] = \mathbb{E}_c \left[ f_W(c) \cdot \left( \mu(\hat{\pi}_c(W)) - \mu(\pi_c^*) \right) \right]. \quad (1)$$

**Lemma 1.** *Loss$(W)$ satisfies the following properties: (1) $\forall W$, Loss$(W) \geq 0$, and Loss$(W^*) = 0$. (2) If Loss$(W) = 0$ then $\forall c \in \mathcal{C}$, the expert policy $\pi_c^*$ is the optimal policy w.r.t. reward $c^T W$.*

To evaluate the loss, the optimal policy $\hat{\pi}_c(W)$ and its features expectations $\mu(\hat{\pi}_c(W))$ must be computed for all contexts. For a specific context, finding $\hat{\pi}_c(W)$ can be solved with standard RL methods such as Value or Policy Iteration. Computing $\mu(\hat{\pi}_c(W))$ is equivalent to policy evaluation (solving linear equations).

The challenge is that Eq. (1) is is not differentiable in $W$. We tackle this problem using two methods for computing descent directions that do not involve differentiation: (i) computing subgradients and (ii) randomly perturbing the loss function. In addition, as the loss is defined in expectation over the contexts, computing it requires to calculate the optimal policy for all contexts. We deal with this issue at the end of Section 3.1. In the special case that $f_W(c)$ is a linear function, Eq. (1) is convex. The following Lemma characterizes Eq. (1) in this case.

**Lemma 2.** *Let $L_{lin}(W) = \mathbb{E}_c \left[ c^T W \cdot \left( \mu(\hat{\pi}_c(W)) - \mu(\pi_c^*) \right) \right]$. We have that: (1) $L_{lin}(W)$ is a convex function. (2) $g(W) = \mathbb{E}_c \left[ c \odot \left( \mu(\hat{\pi}_c(W)) - \mu(\pi_c^*) \right) \right]$ is a sub gradient of $L_{lin}(W)$. (3) $L_{lin}$ is a Lipschitz continuous function, with Lipschitz constant $L = \frac{2}{1-\gamma}$ w.r.t. $\|\cdot\|_\infty$ and $L = \frac{2\sqrt{dk}}{1-\gamma}$ w.r.t. $\|\cdot\|_2$.*

A technical proof (by definition) is provided in the supplementary material. Note that $g(W) \in \mathbb{R}^{d \times k}$; we will sometimes refer to it as a matrix and sometimes as a flattened vector, no confusion will arise.

**Remark 1.** *The Lipschitz of $L_{Lin}(W)$ is related to the simulation lemma (Kearns & Singh, 2002); a small change in the reward results in a small change in the optimal value.*

**Remark 2.** *As $g(W)$ is a subgradient of Loss$(W)$, it can be used to back-propagate DNNs. Clearly, we cannot guarantee convexity (hence no theoretical guarantees), but we can design Loss$(W)$ to be Lipschitz continuous in $W$ using the methods presented in Cisse et al. (2017); Arjovsky et al. (2017).*

**Remark 3.** *The subgradient $g(W)$ is given in expectation over contexts, and in expectation over trajectories (feature expectations). We will later see how to replace it with an unbiased estimate, which can be computed by observing a single expert trajectory for a single context.*

### 3.1  MIRRORED DESCENT FOR COIRL

Lemma 2 identifies $L_{Lin}(W)$ as a convex function and provides a method to compute its subgradients. A standard method for minimizing a convex function over a convex set is the subgradient projection algorithm (Bertsekas, 1997): $w_{t+1} = \text{Proj}_\mathcal{W}\{w_t - \alpha_t g(w_t)\}$, where $f(w_t)$ is a convex function, $g(w_t)$ is a subgradient of $f(w_t)$, and $\alpha_t$ the learning rate. $\mathcal{W}$ is a convex set, and specifically, we consider the $\ell_2$ ball (Abbeel & Ng, 2004) and the simplex (Syed & Schapire, 2008)[1]. We focus on

---

[1]Scaling of the reward by a constant does not affect the resulting policy, thus, these sets are not restricting.

---

**Algorithm 1** MDA for COIRL

   **input:** a convex set $\mathcal{W}$, $T$ number of iterations
   **initialize** $w_1 \in \mathcal{W}$
   **for** $t = 1, \ldots, T$ **do**
      Observe $c, \mu(\pi_c^*)$
      Compute $\hat{\pi}_c(W), \mu(\hat{\pi}_c(W))$
      Compute $g_t$ according to Lemma 2
      **if** PSGD **then**
         $\alpha_t = (1 - \gamma)\sqrt{\frac{1}{2dkt}}$
         $w_{t+1} = w_t - \alpha_t g_t$
         **if** $\|w_{t+1}\| > 1$ **then**
            $w_{t+1} = w_{t+1} / \|w_{t+1}\|_2$
      **else if** Exponential weights **then**
         $\alpha_t = (1 - \gamma)\sqrt{\frac{\log(dk)}{2t}}$
         **for** $i = 1, \ldots, dk$ **do**
            $w_{t+1}(i) = w_t(i) \exp(-\alpha_t g_t(i))$
         $w_{t+1} = w_{t+1} / \sum_i w_{t+1}(i)$
   **return** $\frac{1}{t} \sum_{t=1}^{T} w_t$

**Algorithm 2** ES for COIRL

   **input:** step sizes $\{\alpha_t\}_{t=1}^{T}$s, noise STD $\sigma$, number of evaluations $m$ and smoothing parameter $\nu > 0$
   **initialize:** $W \in \mathbb{R}^k$
   **for** $t = 1, \ldots, T$ **do**
      Observe $c, \mu(\pi_c^*)$
      **for** $j = 1, \ldots, m$ **do**
         $u_j \sim \mathcal{N}^k(0, \sigma^2)$
         $\text{Loss}_j(W) = \text{Loss}\left(W + \frac{u_j}{||u_j||}\nu\right)$
      $d_{\text{Loss}(W)} = \sum_{j=1}^{m} \text{Loss}_j(W)\frac{u_j}{||u_j||}\nu$
      **If** $\text{Loss}(W - \frac{\alpha_t}{b} d_{\text{Loss}(W)}) < \text{Loss}(W)$
         **then** $W = W - \frac{\alpha_t}{m\sigma} d_{\text{Loss}(W)}$
   **return** $W$

---

a generalization of the subgradient projection algorithm that is called the mirror descent algorithm (Nemirovsky & Yudin, 1983, MDA): $w_{t+1} = \arg\min_{w \in \mathcal{W}} \left\{ w \cdot \nabla_f(w_t) + \frac{1}{\alpha_t} D_\psi(w, w_t) \right\}$, where $D_\psi(w, w_t)$ is a Bregman distance[2], associated with a strongly convex function $\psi$. The following theorem characterizes the convergence rate of MDA.

**Theorem 1** (Convergence rate of MDA). *Let $\psi$ be a $\sigma$-strongly convex function on $\mathcal{W}$ w.r.t. $\|\cdot\|$, and let $D^2 = \sup_{w_1, w_2 \in \mathcal{W}} D_\psi(w_1, w_2)$. Let $f$ be convex and $L$-Lipschitz continuous w.r.t. $\|\cdot\|$. Then, MDA with $\alpha_t = \frac{D}{L}\sqrt{\frac{2\sigma}{t}}$ satisfies:* $f\left(\frac{1}{T}\sum_{s=1}^{T} x_s\right) - f(x^*) \leq DL\sqrt{\frac{2}{\sigma T}}$ .

We refer the reader to Beck & Teboulle (2003) and Bubeck (2015) for the proof. Next, we provide two MDA instances (see, for example Beck & Teboulle (2003) for derivation) and analyze them for COIRL.

**Projected subgradient descent (PSGD):** Let $\mathcal{W}$ be an $\ell_2$ ball with radius 1. Fix $||\cdot||_2$, and $\psi(w) = \frac{1}{2}||w||_2^2$. $\psi$ is strongly convex w.r.t. $||\cdot||_2$ with $\sigma = 1$. The associated Bregman divergence is given by $D_\psi(w_1, w_2) = 0.5||w_1 - w_2||_2^2$. Thus, mirror descent is equivalent to PSGD. $D^2 = \max_{w_1, w_2 \in \mathcal{W}} D_\psi(w_1, w_2) \leq 1$, and according to Lemma 2, $L = \frac{2\sqrt{dk}}{1-\gamma}$. Thus, we have that after $T$ iterations $L_{\text{lin}}\left(\frac{1}{T}\sum_{t=1}^{T} w_t\right) - L_{\text{lin}}(w^*) \leq \mathcal{O}\left(\frac{\sqrt{dk}}{(1-\gamma)\sqrt{T}}\right)$.

**Exponential Weights (EW):** Let $\mathcal{W}$ be the standard $dk - 1$ dimensional simplex. Let $\psi(w) = \sum_i w(i) \log(w(i))$. $\psi$ is strongly convex w.r.t. $||\cdot||_1$ with $\sigma = 1$. We get that the associated Bregman divergence is given by $D_\psi(w_1, w_2) = \sum_i w_1(i) \log(\frac{w_1(i)}{w_2(i)})$, also known as the Kullback-Leibler divergence. In addition, $D^2 = \max_{x, y \in \mathcal{W}} D_\psi(w_1, w_2) \leq \log(dk)$ and according to Lemma 2, $L = \frac{2}{1-\gamma}$. Furthermore, the projection onto the simplex w.r.t. to this distance amounts to a simple renormalization $w \leftarrow w/||w||_1$. Thus, we get that MDA is equivalent to the exponential weights algorithm and $L_{\text{lin}}\left(\frac{1}{T}\sum_{t=1}^{T} w_t\right) - L_{\text{lin}}(w^*) \leq \mathcal{O}\left(\frac{\sqrt{\log(dk)}}{(1-\gamma)\sqrt{T}}\right)$.

**Practical MDA:** One of the "miracles" of MDA is its robustness to noise. If we replace $g_t$ with an unbiased estimate $\tilde{g}_t$, such that $\mathbb{E}\tilde{g}_t = g_t$ and $\mathbb{E}\|\tilde{g}_t\| \leq L$, we obtain the same convergence results as in Lemma 2 (Robbins & Monro, 1951) (see, for example, (Bubeck, 2015, Theorem 6.1)). Such an unbiased estimate can be obtained in the following manner: (i) sample a context $c_t$, (ii) compute

---

[2]We refer the reader to Appendix C for definitions of the Bregman distance, the dual norm, etc.

$\mu(\pi^*_{c^T_t w_t})$, (iii) observe a single expert demonstration $\tau^E_i = \{s^i_0, a_0, s^i_1, a_1, \ldots, \}$, where $a_i$ is chosen by the expert policy $\pi^*_{c^T_t w^*}$ (iv) let $\hat{\mu}_i = \sum_{t \in [0,\ldots,|\tau^E_i|-1]} \gamma^t \phi(s^i_t)$ be the accumulated discounted features across the trajectory such that $\mathbb{E}\hat{\mu}_i = \mu(\pi^*_c)$.

The challenge is, that for $\hat{\mu}_i$ to be an unbiased estimate of $\mu(\pi^*_{c^T_t w^*})$, $\tau^E_i$ needs to be of infinite length. There are two ways in which we can tackle this issue. We can either (1) execute the expert trajectory online, and terminate it at each time step with probability $1 - \gamma$ (as in (Kakade & Langford, 2002)), or (2) execute a trajectory of length $H = \frac{1}{1-\gamma} \log(1/\epsilon_H)$. The issue with the first approach is that since the trajectory length is unbounded, the estimate $\hat{\mu}_i$ cannot be shown to concentrate to $\mu(\pi^*_c)$ via Hoeffding type inequalities. Nevertheless, it is possible to obtain a concentration inequality using the fact that the length of each trajectory is bounded in high probability (similar to Zahavy et al.). The second approach can only guarantee that $\|g_t - \mathbb{E}\tilde{g}_t\| \leq \epsilon_H$ (Syed & Schapire, 2008). Therefore, using the robustness of MDA to adversarial noise (Zinkevich, 2003), we get that MDA converges with an additional error of $\epsilon_H$, i.e., $L_{\text{lin}}\left(\frac{1}{T}\sum_{t=1}^T w_t\right) - L_{\text{lin}}(w^*) \leq \mathcal{O}\left(\frac{1}{\sqrt{T}}\right) + \epsilon_H$. While this sampling mechanism comes with the cost of a controlled bias, usually it is more practical, in particular when the trajectories are given as a set demonstrations (offline data).

## 3.2 EVOLUTION STRATEGIES FOR COIRL

To minimize Eq. (1), we also design a derivative free algorithm (Algorithm 2) that is based on *Evolution Strategies* (Salimans et al., 2017, ES). For convex optimization problems, ES is a gradient-free descent method that is based on computing finite differences (Nesterov & Spokoiny, 2017), whose sample complexity is provided below in Theorem 2. The Theorem is given in terms of the Lipschitz constant, which is upper bounded by $\frac{2\sqrt{dk}}{1-\gamma}$ (Section 3.1). While this approach has looser upper-bound guarantees compared to MDA (Theorem 1), Nesterov & Spokoiny (2017) observed that in practice, it often outperforms subgradient based methods. Thus, we test this method empirically and compare it with the subgradient method (Section 3.1). ES is also known to perform well in practice, even with nonconvex objectives. Specifically, Salimans et al. (2017) has shown that ES can be used to optimize the parameters of a DNN to solve challenging high dimensional RL tasks like playing Atari.

**Theorem 2** (ES Convergence Rate (Nesterov & Spokoiny, 2017)). *Let $L_{lin}(W)$ be a non-smooth convex function with Lipschitz constant $L$, such that $\|x_0 - x^*\| \leq D$, step size of $\alpha_t = \frac{D}{(dk+4)\sqrt{T+1}L}$ and $\nu \leq \frac{\epsilon}{2L\sqrt{dk}}$ then in $T = \frac{4(dk+4)^2 D^2 L^2}{\epsilon^2}$ ES finds a solution which is bounded by $\mathbb{E}_{U_{T-1}}[L_{lin}(\hat{x}_T)] - L_{lin}(x^*) \leq \epsilon$, where $U_T = \{u_0, \ldots, u_T\}$ denotes the random variables of the algorithm up to time $T$ and $\hat{x}_T = \arg\min_{t=1,\ldots,T} L_{lin}(x_t)$.*

## 3.3 ELLIPSOID ALGORITHMS FOR COIRL

The final algorithm we consider is an ellipsoid method, introduced to the IRL setting by Amin et al. (2017). In this section we extend it to the contextual setting, specifically, we focus on finding a linear mapping $W$ and further assume that $\mathcal{W} = \{W : \|W\|_\infty \leq 1\}$, and that $W^* \in \mathcal{W}$.

The algorithm maintains an ellipsoid-shaped feasibility set that contains $W^*$. At any step, the current estimation $W_t$ of $W^*$ is defined as the center of the ellipsoid, and the agent acts optimally w.r.t. this estimation. If the agent performs sub-

---

**Algorithm 3** Ellipsoid algorithm for COIRL

**Initialize:** $\Theta_0 \leftarrow B_\infty(0,1) = \{x \in \mathbb{R}^{d \cdot k} : \|x\|_\infty \leq 1\}$
$\Theta_1 \leftarrow \text{MVEE}(\Theta_0)$
**for** $t = 1, 2, \ldots$ **do**
    Observe $c_t$, let $\underline{W}_t$ be the center of $\Theta_t$
    Play episode using $\hat{\pi}_t = \arg\max_\pi V^\pi_{c^T_t W_t}$
    **if** $V^*_{c^T_t W^*} - V^{\hat{\pi}_t}_{c^T_t W^*} > \epsilon$ **then**
        $\mu(\pi^*_{c_t})$ is revealed
        Let $a_t = c_t \odot \left(\mu(\pi^*_{c_t}) - \mu(\hat{\pi}_t)\right)$
        $\Theta_{t+1} \leftarrow \text{MVEE}\left(\{\theta \in \Theta_t : \theta^T a_t \geq \underline{W}^T_t a_t\}\right)$
    **else**
        $\Theta_{t+1} \leftarrow \Theta_t$

---

optimally, the expert provides a demonstration in the form of the optimal feature expectations for $c_t$, $\mu(\pi^*_{c_t})$. The feature expectations are used to generate a linear constraint (hyperplane) on the ellipsoid that is crossing its center. Under this constraint, we construct a new feasibility set that is half of the

previous ellipsoid, and still contains $W^*$. For the algorithm to proceed, we compute a new ellipsoid that is the minimum volume enclosing ellipsoid (MVEE) around this "half-ellipsoid" [3]. These updates are guaranteed to gradually reduce the volume of the ellipsoid (a well-known result (Boyd & Barratt, 1991)) until its center is a mapping which induces $\epsilon$-optimal policies. Theorem 3 shows that this algorithm achieves a polynomial upper bound on the number of sub-optimal time-steps. Finally, note that in Algorithm 3 we use an underline notation to denote a "flattening" operator for matrices, and $\odot$ to denote a composition of an outer product and the flattening operator. The proofs in this section are provided in the supplementary material, and are adapted from (Amin et al., 2017).

**Theorem 3.** *In the linear setting where $R_c^*(s) = c^T W^* \phi(s)$, for an agent acting according to Algorithm 1, the number of rounds in which the agent is not $\epsilon$-optimal is $\mathcal{O}(d^2 k^2 \log(\frac{dk}{(1-\gamma)\epsilon}))$.*

**Remark 4.** *Note that the ellipsoid method presents a new learning framework, where demonstrations are only provided when the agent performs sub-optimally. Thus, the theoretical results in this section cannot be directly compared with those of the descent methods. We further discuss this in the experiments and discussion sections.*

**Remark 5.** *The ellipsoid method does not require a distribution over contexts - an adversary may choose them. MDA can also be easily extended to the adversarial setting via known regret bounds on online MDA (Hazan, 2016).*

**Practical ellipsoid algorithm:** In many real-world scenarios, the expert cannot evaluate the value of the agent's policy and cannot provide its policy or feature expectations. To address these issues, we follow Amin et al. (2017) and consider a relaxed approach, in which the expert evaluates each of the individual actions performed by the agent rather than its policy, and provides finite rollouts instead of a policy or feature expectations (see the supplementary material (Algorithm 4) for pseudo code). We define the expert criterion for providing a demonstration to be $Q_{c_t^T W^*}^*(s,a) + \epsilon < V_{c_t^T W^*}^*(s)$ for each state-action pair $(s,a)$ in the agent's trajectory.

*Near-optimal experts:* In addition, we relax the optimality requirement of the expert and instead assume that, for each context $c_t$, the expert acts optimally w.r.t. $W_t^*$ which is close to $W^*$; the expert also evaluates the agent w.r.t. this mapping. This allows the agent to learn from different experts, and from non-stationary experts whose judgment and performance slightly vary over time. If a sub-optimal action w.r.t. $W_t^*$ is played at state $s$, the expert provides a roll-out of $H$ steps from $s$ to the agent. As this roll-out is a sample of the optimal policy w.r.t. $W_t^*$, we aggregate $n$ examples to assure that with high probability, the linear constraint that we use in the ellipsoid algorithm does not exclude $W^*$ from the feasibility set. Note that these batches may be constructed across different contexts, different experts, and different states from which the demonstrations start. Theorem 4 below upper bounds the number of sub-optimal actions that Algorithm 4 chooses.[4]

**Theorem 4.** *For an agent acting according to Algorithm 4 , with probability of at least $1 - \delta$, for $H = \lceil \frac{1}{1-\gamma} \log(\frac{8k}{(1-\gamma)\epsilon}) \rceil$ and $n = \lceil \frac{512k^2}{(1-\gamma)^2\epsilon^2} \log(4dk(dk+1)\log(\frac{16k\sqrt{dk}}{(1-\gamma)\epsilon})/\delta) \rceil$, if $\forall t$ : $\underline{W_t^*} \in B_\infty(\underline{W^*}, \frac{(1-\gamma)\epsilon}{8k}) \cap \Theta_0$ the number of rounds in which a sub-optimal action is played is $\mathcal{O}\left( \frac{d^2 k^4}{(1-\gamma)^2\epsilon^2} \log\left( \frac{dk}{(1-\gamma)\delta\epsilon} \log(\frac{dk}{(1-\gamma)\epsilon}) \right) \right).$*

## 4 EXPERIMENTS

The simulations in this section include two domains: (1) an autonomous driving simulation (Abbeel & Ng, 2004), that we adapted to the contextual setup and (2) a medical treatment regime, constructed from a data set of expert (clinician) trajectories for treating patients with sepsis[5]. In each of these domains we compare the algorithms in two setups: the ellipsoid learning framework and an offline framework. All the results are averaged across 10 random seeds in Section 4.1 and 5 seeds in Section 4.2 (we report the mean and the standard deviation). Due to space considerations we present the simulations in the ellipsoid framework only for the car domain, and the simulations in the offline framework only in the dynamic treatment regime. Complementary simulations can be found in the supplementary material.

---

[3]This procedure follows a sequence of linear algebra operations which we explain in the appendix.

[4]MDA also works with near optimal experts due to the robustness of MDA. The analysis of this case is identical to the analysis of biased trajectories, as we discuss in the end of Section 3.1.

[5]The data, code and implementation of our algorithms can be found in github.com/CIRLMDP/CIRL.

## 4.1 DRIVING SIMULATION – THE ELLIPSOID FRAMEWORK

In the **ellipsoid framework**, an expert evaluates the agent policy. If the agent's policy is $\epsilon$ sub-optimal, the expert provides the agent its **feature expectations**; otherwise, no demonstration is given. The algorithm performs learning in between demonstrations. This setup enables a proper comparison with the ellipsoid algorithm, which requires the additional expert supervision. We measure performance w.r.t. the following criteria: (1) # demonstrations – the amount of contexts on which each algorithm requested an expert demonstration (y-axis) as a function of time, i.e., the total number of contexts (x-axis). (2) Value – the difference in value, between the agent policy and the expert policy w.r.t. the true reward mapping, i.e., $\sum_{c \in C_{\text{test}}} f_{W^*}(c) \cdot \left( \mu(\hat{\pi}_c(W)) - \mu(\pi_c^*) \right)$, where $C_{\text{test}}$ is a holdout (test) set of contexts. The x-axis measures the amount of demonstrations given.

**Setup.** This domain simulates a three-lane highway with two visible cars - cars A and B (illustration provided in the appendix). The agent, controlling car A, can drive both on the highway and off-road. Car B drives on a fixed lane, at a slower speed than car A. Upon leaving the frame, car B is replaced by a new car, appearing in a random lane at the top of the screen. The feature vector $\phi(s)$ is composed of 3 features: (1) a speed feature, (2) a collision feature, which is valued 0 in case of a collision and 0.5 otherwise, and (3) an off-road feature, which is 0.5 if the car is on the road and 0 otherwise.

In this task, the context vector implies different priorities for the agent; should it prefer speed or safety? Is going off-road to avoid collisions a valid option? For example, an ambulance will prioritize speed and may allow going off-road as long as it goes fast and avoids collisions, while a bus will prioritize avoiding both collisions and off-road driving as safety is its primary concern. To demonstrate the effectiveness of our solutions, the mapping $f : C \mapsto [-1, 1]^k$ is constructed in a way that induces different behaviors for different contexts, making generalization a challenging task. We provide additional details on the domain as well as the hyper parameter selection in the appendix.

**Linear:** The optimal behavior is defined using a linear mapping $W^*$. In this setting, all three approaches obtain competitive results, in terms of generalization, although the ES is capable of obtaining these results faster, as is seen through the regret and number of required demonstrations.

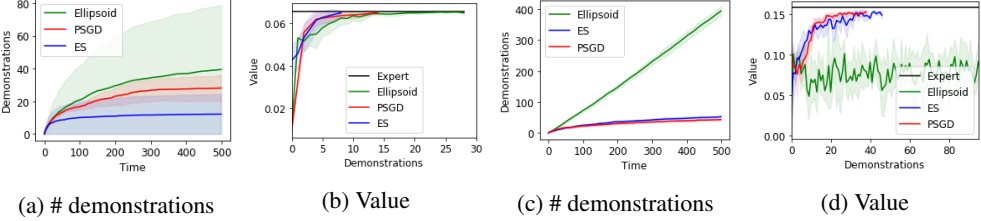

(a) # demonstrations      (b) Value      (c) # demonstrations      (d) Value

Figure 2: Experimental results in the autonomous driving simulation with a **linear** mapping (a & b) and a **nonlinear** mapping (c & d)

**Nonlinear:** For the nonlinear task, we consider two reward coefficient vectors $r_1$ and $r_2$, and define the mapping by $f^*(c) = r_1$ if $||c||_\infty \geq 0.55$, and $r_2$ otherwise - an illustration is provided in the appendix. In order to learn the nonlinear mapping, we represent $f_W(c)$ using a DNN, a multi-layered perceptron, which maps from context to reward vector. DNNs have proven to be capable of extracting meaningful features from complex high-dimensional data, e.g., images - in these scenarios, the linear assumption no longer holds, yet DNNs often overcome such issues. In this setting, the superiority of the descent methods rises; as the linear assumption in the ellipsoid algorithm is not met, it fails to generalize and keeps requiring new demonstrations. We believe these results to be crucial when considering real-life applications, in which the problem is not necessarily linear. Such cases highlight the strength of the descent methods, which, as Fig. 2 shows, are capable of scaling to nonlinear high dimensional mappings.

## 4.2 DYNAMIC TREATMENT REGIME – THE OFFLINE FRAMEWORK

In the **offline framework,** we focus on the ability to learn from previously collected data. A data set of previously collected trajectories is given, such that a single trajectory of finite length is observed for each context and no context is observed more than once. We measure performance w.r.t. the following

criteria: (1) Value – as in the ellipsoid framework above, but here the x-axis corresponds to the amount of iterations. Each iteration corresponds to a single subgradient step, where the subgradient is computed from a mini batch of 10 contexts. (2) Loss – as in Eq. (1). (3) Accuracy % – the percent of actions on which the expert and the agent agree on. All these criteria are evaluated on a holdout set.

**Setup.** In the dynamic treatment regime, there is a clinician which acts to improve a sick patient's medical condition. The context (static information) represents patient features, which do not change during treatment, such as age and gender. The state summarizes the dynamic measurements of the patient, e.g., blood pressure and EEG readouts. The actions are the forms of intervention a clinician may take, including combinations of various treatments provided in parallel. Dynamic treatment regimes are particularly useful for managing chronic disorders and fit well into the broader paradigm of personalized medicine (Komorowski et al., 2018; Prasad et al., 2017).

The agent needs to choose the right treatment for a patient that is diagnosed with sepsis. We use the MIMIC-III data set (Johnson et al., 2016) and follow the data processing steps that were taken in Jeter et al. (2019). As performing off-policy evaluation is not possible using this data set, due to it not satisfying basic requirements (Gottesman et al., 2018; 2019), we designed a simulator of a CMDP. The simulator is based on this data set; a complete overview and explanation on how it was created is provided in the appendix. The mapping $W^*$ is linear, $W^* \in \mathbb{R}^{8 \times 42}$, which we constructed from the data. In the simulator, the expert acts optimally w.r.t. this $W^*$.

Specifically, when treating a sepsis patient, the clinician has several decisions to make, such as whether or not to provide a patient with vasopressors, drugs which are commonly provided to restore and maintain blood pressure in patients with sepsis. However, what is regarded as healthy blood pressure differs based on the age and weight of the patient (Wesselink et al., 2018). In our setting, $W$ captures this information - as it maps from contextual (e.g., age) and dynamic information (e.g., blood pressure) to reward.

**Results.** Fig. 3 presents the ability of the descent methods to generalize to unseen contexts by learning from offline data (without supervision). The data is composed of a set of *trajectories*, i.e., offline data, that were collected from experts (clinicians treating patients). In each iteration, we sample a mini-batch of 10 contexts, i.i.d, from the context distribution. For each context, there is a corresponding expert trajectory of length $H = 40$. Performance is measured on a holdout set of 300 contexts (that are sampled from the same context distribution) according to Theorem 1. We can see that both ES and PSGD attain near-optimal performance using only previously collected expert trajectories.

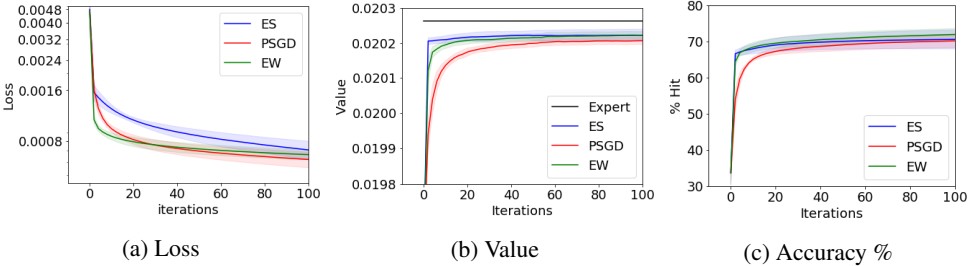

(a) Loss          (b) Value          (c) Accuracy %

Figure 3: Experimental results in the dynamic treatment regime with a **linear** mapping

Looking at Fig. 3a, we can see that all the algorithms manage to minimize the loss to roughly the same error. The small bias is explained by the fact that we use truncated trajectories (as we discussed in the practical MDA paragraph) where in the ellipsoid framework experiments we used feature expectations. We can also see that minimizing the loss leads to policies that attain $\epsilon-$optimal value w.r.t. the true reward Fig. 3b. Finally, in Fig. 3 we can see that all the algorithms reach around 70% accuracy with the expert policy. We emphasize here that 100% accuracy should not be expected for two reasons: (i) different policies may have the same feature expectations (hence the same value) but make different decisions (ii) there exists reward for which there is more than one optimal policy. Nevertheless, Fig. 3 suggests that accuracy is correlated with minimizing the COIRL loss (Eq. (1)).

Finally, we present results in the **nonlinear** setting. Here, there is a non-linear function of the context that determines which one of the two reward coefficient vectors is used, i.e., $f^*(c) = r_1$ if age $>$ 0.1, and $r_2$ otherwise. Where age refers to the normalized age of the patient, which is an element

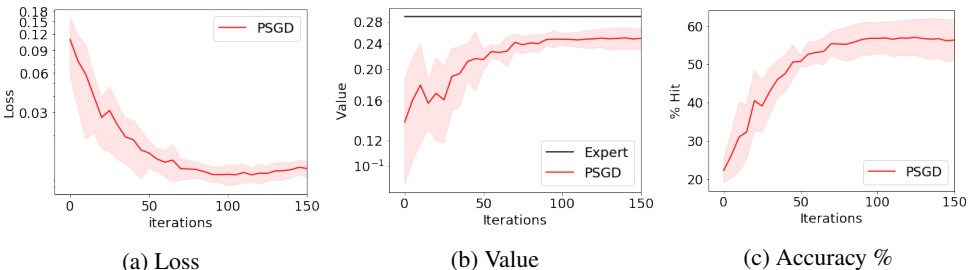

(a) Loss                    (b) Value                    (c) Accuracy %

Figure 4: Experimental results in the dynamic treatment regime with a **non-linear** mapping

of the context vector. We use a DNN to learn the mapping and follow Section 3.1 (PSGD). As seen in Fig. 4, the PSGD algorithm minimizes the loss and achieves a value that is close to that of the expert. In addition, similarly to Fig. 7, accuracy and performance do not necessarily correlate one to another.

## 5 RELATED WORK

We begin with a short discussion on contextual policies, i.e., a policy that is a function of both the state and the context. While there is empirical evidence that learning such a policy may perform well in practice (e.g., Xu et al. (2018); Fu et al. (2019)), from a theoretical point of view, there exist hardness results in this setting. Specifically, given an MDP with $k + 1$ states, there is a reduction from the training problem of the union of $k$ hyperplanes to the policy (see appendix E for proof).

Alternatively, one may consider applying an AL algorithm on a single, large MDP that includes all the states and the contexts. For a concrete example, consider a reduction from the CMDP model to a large MDP where each state is expanded by concatenating the context to it. The new states are $s' = (s, c)$, and the new features are $\phi(s') = c \odot \phi(s)$. Generally speaking, applying an AL algorithm to this large MDP will give the same scalability and sample complexity as COIRL. However, as the large MDP has $|\mathcal{S}'| = |\mathcal{C}||\mathcal{S}|$ states, computing the optimal policy in each iteration of the algorithm will require at least $|C|$ times more time.

To illustrate this problem we conducted a simple grid world experiment on a $3 \times 4$ grid world MDP (with 12 states) and one-hot features ($\phi(s_i) = e_i \in \mathbb{R}^{12}$). The dynamics are deterministic, and the actions correspond to going up, down, left and right (with cyclic transitions on the borders). The contexts correspond to "preferences" on the grid; mathematically, each context is sampled from a uniform distribution over the simplex. The mapping $W^*$ is set to be $I^{12 \cdot 12}$, and for AL, we let $w^*$ to be a flattened version of $W^*$.

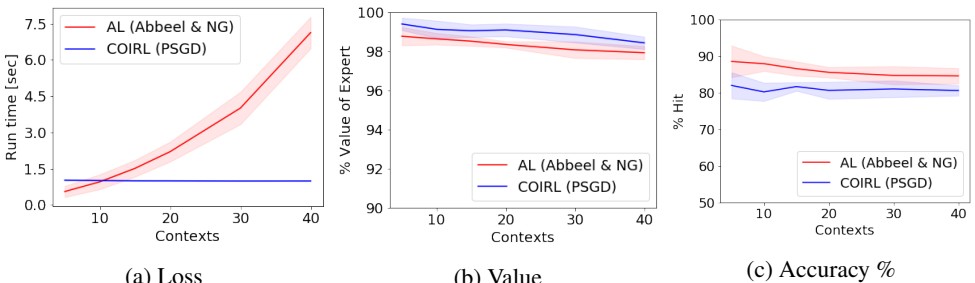

(a) Loss                    (b) Value                    (c) Accuracy %

Figure 5: Comparison between COIRL and AL on a large MDP

We compare the performance of PSGD with the projection algorithm of (Abbeel & Ng, 2004) in Fig. 5. We measure performance by three metrics: run-time, value, and accuracy. Inspecting the results, we can see that AL in the large MDP requires significantly more time to run as the number of contexts grows, while the run time of PSGD (COIRL) is not affected by the number of contexts. We can also see that both methods achieve roughly the same performance: COIRL performs slightly better in terms of value while AL performs slightly better in terms of accuracy. To conclude, AL on a large MDP does not scale to problems with large context spaces. In addition, this construction is

only possible when there is a finite number of context and does not provide generalization results. We avoided all of these issues in the COIRL framework.

## 6 SUMMARY AND DISCUSSION

In this work, we formulated and studied the COIRL problem. We presented two types of algorithms to solve it: **(1)** cutting plane methods (ellipsoid) and **(2)** iterative descent approaches (MDA and ES). We summarize the theoretical guarantees of the different algorithms in Table 1.

We can see that the iterative descent approaches have better dependence in $dk$ than the ellipsoid method, i.e., they **scale** better with the dimensions of the problem. In particular, the EW algorithm has a logarithmic dependence in $dk$, which makes it computationally comparable to standard IRL/AL (on a single, noncontextual MDP). In addition, the iterative methods extend naturally to the more general scenario where the mapping from contexts to rewards is not linear, and $f_W$ is modeled as a DNN. As Sutton (2019) puts it: "The biggest lesson that can be read from 70 years of AI research is that general methods that leverage computation are ultimately the most effective, and by a large margin".

The ellipsoid method has better **sample complexity** (as a function of $\epsilon$) than the descent methods in the deterministic setting. However, both methods attain the same complexity in the more realistic, stochastic setting. Our empirical findings suggest that the iterative methods always outperform the ellipsoid algorithm. Among these methods, we found the ES method to perform better than the MDA method. Similar findings were reported in (Nesterov & Spokoiny, 2017) for other convex problems.

The iterative methods have another advantage over the ellipsoid method – they can learn from previously collected demonstrations (i.e., offline learning). The ellipsoid framework, on the other hand, requires expert **supervision** throughout the entire learning process.

Finally, an attractive property of the ellipsoid learning framework is its **safety**, i.e., an IRL algorithm that is being supervised by an expert will never perform sub-optimally. In each step, either that the agent performs $\epsilon$-optimally or that the expert acts on its behalf (provides a demonstration). This property is appealing in mission-critical domains where errors have a high cost; for instance, in health-care, a failure may result in a loss of lives. In the experimental section, we have seen that we can use this learning framework for the iterative methods as well while enjoying improved efficiency.

| | | **Scalability** | | **Sample Complexity** | | **Extension to DNNs** |
|---|---|---|---|---|---|---|
| | | Deterministic | Stochastic | Deterministic | Stochastic | |
| MDA | PSGD | $\mathcal{O}(dk)$ | | $\mathcal{O}\left(\frac{1}{\epsilon^2}\right)$ | $\mathcal{O}\left(\frac{1}{\epsilon^2}\right)$ | ✓ |
| | EW | $\mathcal{O}(\log dk)$ | | | | ✗ |
| ES | | $\mathcal{O}(dk)$ | $\mathcal{O}(d^2k^2)$ | | | ✓ |
| Ellipsoid | | $\mathcal{O}(d^2k^2)$ | $\mathcal{O}(d^2k^4)$ | $\mathcal{O}\left(\frac{1}{\log(1/\epsilon)}\right)$ | | ✗ |

Table 1: Comparison between various approaches.

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

# Appendices

## CONTENTS

# A    COMPLEMENTARY SIMULATIONS

## A.1    AUTONOMOUS DRIVING SIMULATION

Similar to Section 4.2, we compare the various methods in the offline framework. We can see that all the algorithms manage to minimize the loss, and achieve near optimal value. We can also see that they achieve high accuracy with the expert policy but not 100%.

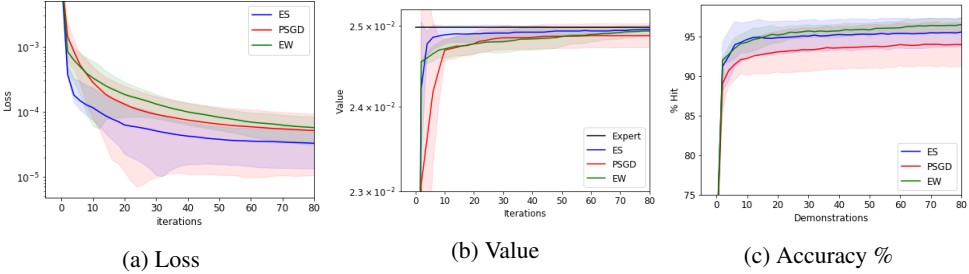

(a) Loss    (b) Value    (c) Accuracy %

Figure 6: Offline framework, autonomous driving simulation, **linear** mapping

## A.2    DYNAMIC TREATMENT REGIME

Similar to Section 4.1, we compare the various methods in the ellipsoid framework . We observe that ES outperforms the ellipsoid method. Additionally, we compare the accuracy, i.e., how often does the policy derived from $W$ match the expert's policy, which is derived from $W^*$. As IRL is *not* a supervised learning problem, we observe that while there is a correlation between the success in the task and the ability to act similarly to the experts policy - this correlation is not strict, in the sense that the agent is capable of finding near-optimal policies with a relatively high miss rate (accuracy of approximately 80%). For more intuition, see the proof of Lemma 1.

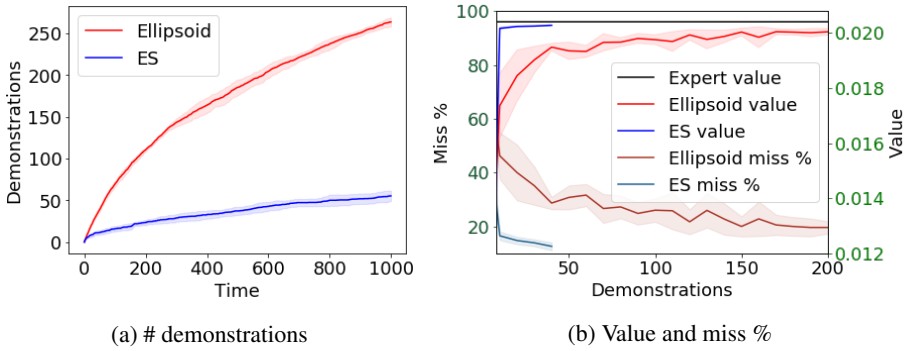

(a) # demonstrations    (b) Value and miss %

Figure 7: Ellipsoid framework, dynamic treatment regime, **linear** mapping

## B  EXPERIMENTAL DETAILS

In this section, we describe the technical details of our experiments, including the hyper-parameters used. To solve MDPs, we use value iteration. Our implementation is based on a stopping condition with a tolerance threshold, $\tau$, such that the algorithm stops if $|V_t - V_{t-1}| < \tau$. In the driving simulation we used $\tau = 10^{-4}$ and in the sepsis treatment we use $\tau = 10^{-3}$.

### B.1  AUTONOMOUS DRIVING SIMULATION



Figure 8: Driving simulator

The environment is modeled as a tabular MDP that consists of 1531 states. The speed is selected once, at the initial state, and is kept constant afterward. The other 1530 states are generated by 17 X-axis positions for the agent's car, 3 available speed values, 3 lanes and 10 Y-axis positions in which car B may reside. During the simulation, the agent controls the steering direction of the car, moving left or right, i.e., two actions.

In these experiments, we define our mappings in a way that induces different behaviours for different contexts, making generalization a more challenging task. Specifically, for the linear setting we use $W^* = \begin{pmatrix} -1 & 0.75 & 0.75 \\ 0.5 & -1 & 1 \\ 0.75 & 1 & -0.75 \end{pmatrix}$, before normalization. For our nonlinear mapping, contexts with $||c||_\infty > 0.55$ are mapped to reward coefficients vector $(1, -1, -0.05)$, otherwise they are mapped to $(-0.01, 1, -1)$, which induce the feature expectations $(9.75, 3.655, 5)$, $(5.25, 5, 2.343)$ respectively. The decision regions for the nonlinear mapping are visualized in Appendix B.1. The contexts are sampled uniformly in the 2-dimensional simplex. We evaluate all algorithms on the same sequences of contexts, and average the results over 20 such sequences. The algorithms were modified to fit the Ellipsoid framework; instead of iterating over the whole data, the algorithms iterate over the given expert feature expectations, one at a time, until convergence, i.e. every timestep a new demonstration is presented.

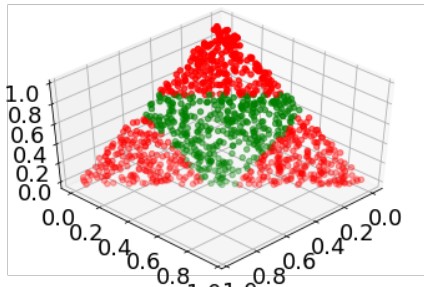

Figure 9: Visualization of nonlinear decision boundaries

**Hyper-parameter selection and adjustments**:

**Ellipsoid Framework**: For the **linear** model the algorithms maintained a $3 \times 3$ matrix to estimate $W^*$. *Ellipsoid:* By definition, the ellipsoid algorithm is hyper-parameter free and does not require tuning.

*PSGD:* The algorithm was executed with with the parameters: $\alpha_0 = 0.3, \alpha_t = 0.9^t \alpha_{t-1}$, and iterated for 40 epochs. An outer decay on the step size was added for faster convergence, the initial $\alpha_0$

becomes $0.94 \cdot \alpha_0$ every time a demonstration is presented. The gradient, $g_t$ is normalized to be $g_t = g_t \frac{g_t}{||g_t||_\infty}$ and the calculated step is taken if: $cW_t\big(\mu(\hat{\pi}_c^t) - \mu(\pi_c^*)\big) > cW_{t+1}\big(\mu(\hat{\pi}_c^{t+1}) - \mu(\pi_c^*)\big)$, where $\hat{\pi}_c^t$ denotes the optimal policy for a context $c$ according to $W_t$.

*ES:* The algorithm was executed with the parameters: $\sigma = 10^{-3}, m = 250, \alpha = 0.1$ with decay rate of 0.95, for 50 iterations which didn't iterate randomly over one the contexts, but rather used the entire training set (all of the observed contexts and expert demonstrations up to the current time-step) for each step. The matrix was normalized according to $|| \cdot ||_2$, and so was the step calculated by the ES algorithm, before it was multiplied by $\alpha$ and applied.

For the **nonlinear setting**, the model used for the ES method was a fully connected DNN, with layers of sizes $15, 10, 5, 3$. The activation function used was the leaky ReLU function, with a parameter of $\alpha = 0.1$. Note that we can't normalize the parameters here as in the linear case; therefore an L2-normalization layer is added to the output. The same parameters were used as in the linear case, except with 120 iterations over the entire training set. They were originally optimized for this model and setting and worked as-is for the linear environment. As we aim to estimate the gradient, a small $\sigma = 10^{-3}$ was used and performed best. The number of points, $m = 250$, was selected as fewer points produced noisy results. The step size, decay rate and the number of iterations were $0.1, 0.96, 120$ respectively, and were selected in a way that produced fast yet accurate convergence of the loss. Note that here the steps were also normalized before application, and the normalization was applied per layer. For PSGD, a similar network was used. Specifically, it had layer sizes $14, 10, 6, 3$, and the same leaky ReLU activation function was used in this network. In parallel to the normalization used for the ES model, here we used batch normalization and gradient clipping. The learning rate was set to $0.1 \cdot 0.98^t$ and 120 iterations were performed. For this result, as with the ES method, the batch consisted of all available training data.

**Offline Framework**: In the offline framework we compute the subgradients using expert trajectories of length $40$, instead of the feature expectations. In this framework at every iteration we sample a mini batch of 10 contexts (from a finite set) and their corresponding trajectories (sampled from the expert policy and dynamics) then taking one descent step according to them. The generalization is measured over 80 holdout contexts that are referred as the test set, where the $W$ that is used to calculate the feature expectations of the agent is fitted to the EW algorithm requirement to be in the $dk - 1$ simplex. The PSGD and the EW algorithms are configured as the theory specifies, where each descent step is calculated from the whole batch. The ES algorithm is applied with the parameters $\sigma = 10^{-3}, m = 500, \alpha = 0.1$ with decay rate $0.95$, for every iteration.

## B.2 DYNAMIC TREATMENT REGIME

The environment we describe in 4.2 simulates a decision-making process for treating sepsis. Sepsis is a life-threatening severe infection, where the treatment applied to a sepsis patient is crucial for saving his life. To create a sepsis treating simulator, we leverage the MIMIC-III data set (Johnson et al., 2016). This data set includes data from hospital electronic databases, social security, and archives from critical care information systems, that had been acquired during routine hospital care. We follow the data processing steps that were taken in Jeter et al. (2019) to extract the relevant data in a form of normalized measurements of sepsis patients during their hospital admission and the treatments that were given to each patient. The measurements include dynamic measures, e.g., heart rate, blood pressure, weight, body temperature, blood analysis standard measures (glucose, albumin, platelets count, minerals, etc.), as well as static measures such as age, gender, re-admission (of the patient), and more.

The processed data from Jeter et al. (2019) consists of 5366 trajectories, each representing the sequential treatment provided by a clinician to a patient. At each time-step, the available information for each patient consists of 8 static measurements and 41 dynamic measurements. In addition, each trajectory contains the reported actions performed by the clinician (the number of fluids and vasopressors given to a patient at each time-step and binned to 25 different values), and there is a mortality signal which indicates whether the patient was alive 90 days after his hospital admission.

In order to create a tabular CMDP from the processed data, we separate the static measurements of each patient and keep them as the context. We cluster the dynamic measurements using K-means (MacQueen et al., 1967). Each cluster is considered a state and the coordinates of the cluster centroids are taken as its features $\phi(s)$. We construct the transition kernel between the clusters using the

empirical transitions in the data given the state and the performed actions. Two states are added to the MDP and the feature vector is extended by 1 element, corresponding to whether or not the patient died within the 90 days following hospital release. This added feature receives a value of 0 on all non-terminal states, a value of $-0.5$ for the state representing the patient's death and 0.5 for the one representing survival. In addition, as the data is limited, not all state-action pairs are available. In order to ensure the agent does not attempt to perform such an action for which the outcome is unknown, we add an additional terminal state. At this state, all features are set to $-1$ to make it clearly distinguishable from all other states in the CMDP.

In our simulator, we used the same structure as the raw data, i.e., we used the same contexts prevalent in the data and the same initial state distribution. Each context is projected onto the simplex and the expert's feature expectations for each context are attained by solving the CMDP. While we focus on a simulator, as it allows us to analyze the performance of the algorithms, our goal is to have a reward structure which is influenced by the data. Hence, we produce $W^*$ by running the ellipsoid algorithm on trajectories obtained from the data. As done in the Autonomous Driving Simulation, the algorithms were modified to fit the Ellipsoid framework.

**Hyper-parameter selection and adjustments**:

**Ellipsoid Framework**: ES algorithm: the same method as in the autonomous driving is applied, with the parameters $\sigma = 10^{-4}, m = 1000, \alpha = 0.25$ with decay rate 0.95, for 80 iterations over the entire training set.

**Offline Framework**: In the offline framework for the **linear setting**, we compute the subgradients using expert trajectories of length 40, instead of the exact feature expectations. At every iteration, we sample a mini batch of 10 contexts (from a finite set) and their corresponding trajectories (sampled from the expert policy and dynamics) and perform a single descent step. Generalization is measured over a set of 300 holdout contexts, referred to as the test set, where $W$ is fitted to be in the $dk - 1$ simplex. The PSGD and the EW algorithms are configured as specified by the theory, where each descent step is calculated from the entire batch. The ES algorithm is applied with the parameters $\sigma = 10^{-4}, m = 1000, \alpha = 0.3$ with decay rate 0.95, for every iteration.

For the **nonlinear setting**, the model used for the ES method was a fully connected DNN, with layers of sizes $24, 42, 42$ which include a bias term. The activation function used was the leaky ReLU function, with a parameter of $\alpha = 0.1$. In this setting we use trajectories of length 80, mini batch of 32 contexts. The PSGD algorithm uses the gradient computed by back-propagation of the loss function value on the DNN, where each descent step is calculated from the entire batch. The algorithm is applied with modified parameters; $\alpha = 0.1$ with decay rate 0.98, for every iteration.

## C    PROOFS FOR SECTION 3

**Definition 1** (Bregman distance). *Let $\psi : \mathcal{W} \to R$ be strongly convex and continuously differential in the interior of $\mathcal{W}$. The Bregman distance is defined by $D_\psi(x, y) = \psi(x) - \psi(y) - (x - y) \cdot \nabla\psi(y)$, where $\psi$ is strongly convex with parameter $\sigma$.*

**Definition 2** (Conjugate function). *The conjugate of a function $\psi(y)$, denoted by $\psi^*(y)$ is $\max_{x \in \mathcal{W}} \{x \cdot y - \psi(x)\}$.*

**Example**: let $\| \cdot \|$ be a norm on $\mathbb{R}^n$. The associated dual norm, denoted $\| \cdot \|_*$, is defined as $\|z\|_* = \sup\{z^\mathsf{T} x \mid \|x\| \leq 1\}$. The dual norm of $\| \cdot \|_2$ is $\| \cdot \|_2$, and the dual norm of $\| \cdot \|_1$ is $\| \cdot \|_\infty$.

Before we begin with the proof of Lemma 2, we make the following observation. By definition, $\hat{\pi}_c(W)$ is the optimal policy w.r.t. $c^T W$; i.e., for any policy $\pi$ we have that

$$c^T W \cdot \mu(\hat{\pi}_c(W)) \geq c^T W \cdot \mu(\pi). \tag{2}$$

*Proof of Lemma 1.*
Fix $W$. For any context $c$, we have that $\mu(\hat{\pi}_c)$ is the optimal policy w.r.t. reward $f_W(c)$, thus, $f_W(c) \cdot \big(\mu(\hat{\pi}_c(W)) - \mu(\pi_c^*)\big) \geq 0$. Therefore we get that $\mathrm{Loss}(W) \geq 0$. For $W^*$, we have that $\mu(\hat{\pi}_c(W)) = \mu(\pi_c^*)$, thus $\mathrm{Loss}(W^*) = 0$.

For the second statement, note that $\mathrm{Loss}(W) = 0$ implies that $\forall c, \;\; f_W(c) \cdot \big(\mu(\hat{\pi}_c(W)) - \mu(\pi_c^*)\big) = 0$. This can happen in one of two cases. (1) $\mu(\hat{\pi}_c(W)) = \mu(\pi_c^*)$, in this case $\pi_c^*, \hat{\pi}_c(W)$ have the same feature expectations. Therefore, they are equivalent in terms of value. (2) $\mu(\hat{\pi}_c(W)) \neq \mu(\pi_c^*)$, but $f_W(c) \cdot \big(\mu(\hat{\pi}_c(W)) - \mu(\pi_c^*)\big) = 0$. In this case, $\pi_c^*, \hat{\pi}_c(W)$ have different feature expectations, but still achieve the same value w.r.t. reward $f_W(c)$. Since $\hat{\pi}_c(W)$ is an optimal policy w.r.t. this reward, so is $\pi_c^*$. $\qquad\square$

*Proof of Lemma 2.*

**1.** We need to show that $\forall W_1, W_2 \in \mathcal{W}, \forall \lambda \in [0, 1]$, we have that

$$L_{\mathrm{lin}}(\lambda W_1 + (1 - \lambda)W_2) \leq \lambda L_{\mathrm{lin}}(W_1) + (1 - \lambda)L_{\mathrm{lin}}(W_2)$$

$$
\begin{aligned}
& L_{\mathrm{lin}}(\lambda W_1 + (1 - \lambda)W_2) \\
&= \mathbb{E}_c\left[c^T(\lambda W_1 + (1 - \lambda)W_2) \cdot \Big(\mu\big(\hat{\pi}_c\big(\lambda W_1 + (1 - \lambda)W_2\big)\big) - \mu(\pi_c^*)\Big)\right] \\
&= \lambda \mathbb{E}_c\left[c^T W_1 \cdot \Big(\mu\big(\hat{\pi}_c\big(\lambda W_1 + (1 - \lambda)W_2\big)\big) - \mu(\pi_c^*)\Big)\right] \\
&\quad + (1 - \lambda)\mathbb{E}_c\left[c^T W_2 \cdot \Big(\mu\big(\hat{\pi}_c\big(\lambda W_1 + (1 - \lambda)W_2\big)\big) - \mu(\pi_c^*)\Big)\right] \\
&\leq \lambda \mathbb{E}_c\left[c^T W_1 \cdot \Big(\mu\big(\hat{\pi}_c(W_1)\big) - \mu(\pi_c^*)\Big)\right] + (1 - \lambda)\mathbb{E}_c\left[c^T W_2 \cdot \Big(\mu\big(\hat{\pi}_c(W_2)\big) - \mu(\pi_c^*)\Big)\right] \\
&= \lambda L_{\mathrm{lin}}(W_1) + (1 - \lambda)L_{\mathrm{lin}}(W_2),
\end{aligned}
$$

where the inequality follows from Eq. (2).

**2.** Fix $z \in \mathcal{W}$. We have that

$$
\begin{aligned}
L_{\mathrm{lin}}(z) &= \mathbb{E}_c\left[c^T z \cdot \Big(\mu(\hat{\pi}_c(z)) - \mu(\pi_c^*)\Big)\right] \\
&\geq \mathbb{E}_c\left[c^T z \cdot \Big(\mu(\hat{\pi}_c(W)) - \mu(\pi_c^*)\Big)\right] \\
&= L_{\mathrm{lin}}(W) + (z - W) \cdot \mathbb{E}_c\left[c \odot \Big(\mu(\hat{\pi}_c(W)) - \mu(\pi_c^*)\Big)\right],
\end{aligned}
$$

where the inequality follows from Eq. (2).

**3.** Recall that a bound on the dual norm of the subgradients implies Lipschitz continuity for convex functions. Thus it is enough to show that $\forall W \in \mathcal{W}, \|g(W)\|_p =$

$\left\| \mathbb{E}_c \left[ c \odot \left( \mu(\hat{\pi}_c(W)) - \mu(\pi_c^*) \right) \right] \right\|_p \leq L$. Let $p = \infty$, we have that

$$
\begin{aligned}
\|g(W)\|_\infty &= \left\| \mathbb{E}_c c \odot \left( \mu(\hat{\pi}_c(W)) - \mu(\pi_c^*) \right) \right\|_\infty \\
&\leq \mathbb{E}_c \| c \odot \left( \mu(\hat{\pi}_c(W)) - \mu(\pi_c^*) \right) \|_\infty \qquad \text{(Jensen inequality)} \\
&\leq \mathbb{E}_c \| c \|_\infty \| \mu(\pi_i) - \mu(\pi_j) \|_\infty \leq \frac{2}{1 - \gamma}.
\end{aligned}
\tag{3}
$$

where in Eq. (3) we used the fact that $\forall \pi$ we have that $\| \mu(\pi) \|_\infty \leq \frac{1}{1-\gamma}$, thus, for any $\pi_i, \pi_j$, $\| \mu(\pi_i) - \mu(\pi_j) \|_\infty \leq \frac{2}{1-\gamma}$

Therefore, $L = \frac{2}{1-\gamma}$ w.r.t. $\|\cdot\|_\infty$. Since $\|\cdot\|_2 \leq \sqrt{dk} \|\cdot\|_\infty$ we get that $L = \frac{2\sqrt{dk}}{1-\gamma}$ w.r.t. $\|\cdot\|_2$ . $\qquad \square$

# D    PROOFS & PSEUDO CODE FOR SECTION 3.3

## D.1    ELLIPSOID ALGORITHM FOR TRAJECTORIES

---

**Algorithm 4** Batch ellipsoid algorithm for COIRL

---

**Initialize:** $\Theta_0 \leftarrow B_\infty(0,1) = \{x \in \mathbb{R}^{d \cdot k} : ||x||_\infty \leq 1\}$
$\Theta_1 \leftarrow \text{MVEE}(\Theta_0)$
$i \leftarrow 0, \bar{Z} \leftarrow 0, \bar{Z}^* \leftarrow 0$
**for** $t = 1, 2, 3, ...$ **do**
    $c_t$ is revealed, Let $\underline{W}_t$ be the center of $\Theta_t$
    Play episode using $\hat{\pi}_t = \arg\max_\pi V^\pi_{c_t^T W_t}$
    $\Theta_{t+1} \leftarrow \Theta_t$
    **if** a sub-optimal action $a$ is played at state $s$ **then**
        Expert provides H-step trajectory $(s_0^E = s, s_1^E, ..., s_H^E)$. Let $\hat{x}_i^{*,H}$ be the H-step sample of
the expert's feature expectations for $\xi_i' = \mathbb{1}_s$: $\hat{x}_i^{*,H} = \sum_{h=0}^H \gamma^h \phi(s_h^E)$
        Let $x_i$ be the agent's feature expectations for $\xi_i'$: $E_{\xi_i',P,\pi_t}[\sum_{h=0}^\infty \gamma^h \phi(s_h)]$
        Denote $z_i = c_t \odot x_i$, $\hat{z}_i^{*,H} = c_t \odot \hat{x}_i^{*,H}$
        $i \leftarrow i + 1, \bar{Z} \leftarrow \bar{Z} + z_i, \bar{Z}^* \leftarrow \bar{Z}^* + \hat{z}_i^{*,H}$
        **if** $i = n$ **then**
            $\Theta_{t+1} \leftarrow \text{MVEE}\left(\left\{\theta \in \Theta_t : \left(\theta - \underline{W}_t\right)^T \cdot (\frac{\bar{Z}^*}{n} - \frac{\bar{Z}}{n}) \geq 0\right\}\right)$
            $i \leftarrow 0, \bar{Z} \leftarrow 0, \bar{Z}^* \leftarrow 0$

---

## D.2    MVEE COMPUTATION

This computation is commonly found in optimization lecture notes and textbooks. First, we define an ellipsoid by $\{x : (x-c)Q^{-1}(x-c) \leq 1\}$ for a vector $c$, the center of the ellipsoid, and an invertible matrix $Q$. Our first task is computing $\Theta_1$- the MVEE for the initial feasibility set $\Theta_0 = B_\infty(0,1) = \{x \in \mathbb{R}^{d \cdot k} : ||x||_\infty \leq 1\}$. The result is of course a sphere around 0: $c_1 = 0, Q_1 = dkI$.

For the update $\Theta_{t+1} \leftarrow \text{MVEE}\left(\left\{\theta \in \Theta_t : \left(\theta - \underline{W}_t\right)^T \cdot a_t \geq 0\right\}\right)$, we define $\tilde{a}_t = \frac{-1}{\sqrt{a_t^T Q_t a_t}} a_t$

and calculate the new ellipsoid by $c_{t+1} = c_t - \frac{1}{dk+1} Q\tilde{a}_t$ , $Q_{t+1} = \frac{d^2 k^2}{d^2 k^2 - 1}(Q_t - \frac{2}{dk+1} Q_t \tilde{a}_t \tilde{a}_t^T Q_t)$.

## D.3    PROOF OF THEOREM 3

For simpler analysis, we define a "flattening" operator, converting a matrix to a vector: $\mathbb{R}^{d \times k} \to \mathbb{R}^{d \cdot k}$ by $\underline{W} = [w_{1,1}, \ldots, w_{1,k}, \ldots, w_{d,1}, \ldots, w_{d,k}]$. We also define the operator $\odot$ to be the composition of the flattening operator and the outer product: $u \odot v = [u_1 v_1, \ldots, u_1 v_k, \ldots, u_d v_1, \ldots, u_d v_k]$. Therefore, the value of policy $\pi$ for context $c$ is given by $V^\pi_{c^T W^*} = c^T W^* \mu(\pi) = \underline{W}^{*T}(c \odot \mu(\pi))$, where $||\underline{W}^*||_\infty \leq 1, ||c \odot \mu(\pi)||_1 \leq \frac{k}{1-\gamma}$.

**Lemma 3** (Boyd & Barratt (1991)). *If $B \subseteq \mathbb{R}^D$ is an ellipsoid with center $w$, and $x \in \mathbb{R}^D \backslash \{0\}$, we define $B^+ = MVEE(\{\theta \in B : (\theta - w)^T x \geq 0\})$, then: $\frac{Vol(B^+)}{Vol(B)} \leq e^{-\frac{1}{2(D+1)}}$ .*

*Proof of Theorem 3.* We prove the theorem by showing that the volume of the ellipsoids $\Theta_t$ for $t = 1, 2, ...$ is bounded from below. In conjunction with Lemma 3, which claims there is a minimal rate of decay in the ellipsoid volume, this shows that the number of times the ellipsoid is updated is polynomially bounded.

We begin by showing that $\underline{W}^*$ always remains in the ellipsoid. We note that in rounds where $V^{\pi^*}_{c_t^T W^*} - V^{\hat{\pi}_t}_{c_t^T W^*} > \epsilon$, we have $\underline{W}^{*T}\left(c_t \odot \left(\mu(\pi^*_{c_t}) - \mu(\hat{\pi}_t)\right)\right) > \epsilon$. In addition, as the agent acts optimally w.r.t. the reward $r_t = c_t^T W_t$, we have that $\underline{W}_t^T\left(c_t \odot \left(\mu(\pi^*_{c_t}) - \mu(\hat{\pi}_t)\right)\right) \leq 0$ . Combining

these observations yield:

$$(\underline{W}^* - \underline{W}_t)^T \cdot \left( c_t \odot \left( \mu(\pi_{c_t}^*) - \mu(\hat{\pi}_t) \right) \right) > \epsilon > 0 \ . \tag{4}$$

This shows that $\underline{W}^*$ is never disqualified when updating $\Theta_t$ . Since $\underline{W}^* \in \Theta_0$ this implies that $\forall t : \underline{W}^* \in \Theta_t$. Now we show that not only $\underline{W}^*$ remains in the ellipsoid, but also a small ball surrounding it. If $\theta$ is disqualified by the algorithm: $(\theta - \underline{W}_t)^T \cdot \left( c_t \odot \left( \mu(\pi_{c_t}^*) - \mu(\hat{\pi}_t) \right) \right) < 0$ . Multiplying this inequality by -1 and adding it to (4) yields: $(\underline{W}^* - \theta)^T \cdot \left( c_t \odot \left( \mu(\pi_{c_t}^*) - \mu(\hat{\pi}_t) \right) \right) > \epsilon.$

We apply Hölder inequality to LHS: $\epsilon < \text{LHS} \leq ||\underline{W}^* - \theta||_\infty \cdot ||\left( c_t \odot \left( \mu(\pi_{c_t}^*) - \mu(\hat{\pi}_t) \right) \right)||_1 \leq \frac{2k}{1-\gamma} ||\underline{W}^* - \theta||_\infty$. Therefore for any disqualified $\theta$: $||\underline{W}^* - \theta||_\infty > \frac{(1-\gamma)\epsilon}{2k}$, thus $B_\infty(\underline{W}^*, \frac{(1-\gamma)\epsilon}{2k})$ is never disqualified. This implies that $\forall t : \text{vol}(\Theta_t) \geq \text{vol}(\Theta_0 \cap B_\infty(\underline{W}^*, \frac{(1-\gamma)\epsilon}{2k})) \geq \text{vol}(B_\infty(\underline{W}^*, \frac{(1-\gamma)\epsilon}{4k}))$. Finally, let $M_T$ be the number of rounds by $T$ in which $V_{c_t^T W^*}^{\pi^*} - V_{c_t^T W^*}^{\hat{\pi}_t} > \epsilon$. Using Lemma 3 we get that: $\frac{M_T}{2(dk+1)} \leq \log\left(\text{vol}(\Theta_1)\right) - \log\left(\text{vol}(\Theta_{T+1})\right) \leq \log\left(\text{vol}\left(\text{MVEE}(B_\infty(0,1))\right)\right) - \log\left(\text{vol}(B_\infty(0, \frac{(1-\gamma)\epsilon}{4k}))\right) \leq \log\left(\text{vol}\left(\text{MVEE}(B_2(0, \sqrt{dk}))\right)\right) - \log\left(\text{vol}(B_2(0, \frac{(1-\gamma)\epsilon}{4k}))\right) \leq \log\left(\left(\frac{4k\sqrt{dk}}{(1-\gamma)\epsilon}\right)^{dk}\right) \leq dk \log \frac{4k\sqrt{dk}}{(1-\gamma)\epsilon}$ . Therefore $M_T \leq 2dk(dk+1) \log \frac{4k\sqrt{dk}}{(1-\gamma)\epsilon} = \mathcal{O}(d^2 k^2 \log(\frac{dk}{(1-\gamma)\epsilon}))$ . $\qquad \square$

### D.4    PROOF OF THEOREM 4

**Lemma 4** (Azuma's inequality)**.** *For a martingale* $\{S_i\}_{i=0}^n$, *if* $|S_i - S_{i-1}| \leq b$ *a.s. for* $i = 1, ..., n$: $P\left(|S_n - S_0| > b\sqrt{2n\log(\frac{2}{\delta})}\right) < \delta$

*Proof of Theorem 4.* We first note that we may assume that for any $t$: $||W^* - W_t||_\infty \leq 2$. If $\underline{W}_t \notin \Theta_0$, we update the ellipsoid by $\Theta_t \leftarrow \text{MVEE}\left(\left\{\theta \in \Theta_t : \left(\theta - \underline{W}_t\right)^T \cdot e_j \lessgtr 0\right\}\right)$ where $e_j$ is the indicator vector of coordinate $j$ in which $\underline{W}_t$ exceeds 1, and the inequality direction depends on the sign of $(\underline{W}_t)_j$. If $\underline{W}_t \notin \Theta_0$ still, this process can be repeated for a finite number of steps until $\underline{W}_t \in \Theta_0$, as the volume of the ellipsoid is bounded from below and each update reduces the volume (Lemma 3). Now we have $\underline{W}_t \in \Theta_0$, implying $||W^* - W_t||_\infty \leq 2$. As no points of $\Theta_0$ are removed this way, this does not affect the correctness of the proof. Similarly, we may assume $||W_t^* - W_t||_\infty \leq 2$ as $W_t^* \in \Theta_0$.

We denote $W_t$ which remains constant for each update in the batch by $W$. We define $t(i)$ the time-steps corresponding to the demonstrations in the batch for $i = 1, ..., n$. We define $z_i^{*,H}$ to be the expected value of $\hat{z}_i^{*,H}$, and $z_i^*$ to be the outer product of $c_{t(i)}$ and the feature expectations of the expert policy for $W_{t(i)}^*, c_{t(i)}, \xi_{t(i)}'$ . We also denote $W_{t(i)}^*$ by $W_i^*$. We bound the following term from below, as in Theorem 3:

$$(\underline{W}^* - \underline{W})^T \cdot (\frac{\bar{Z}^*}{n} - \frac{\bar{Z}}{n}) = \frac{1}{n} \sum_{i=1}^n (\underline{W}^* - \underline{W})^T \cdot (\hat{z}_i^{*,H} - z_i) =$$

$$\frac{1}{n} \sum_{i=1}^n (\underline{W}^* - \underline{W})^T \cdot (z_i^* - z_i) + \frac{1}{n} \sum_{i=1}^n (\underline{W}^* - \underline{W})^T \cdot (z_i^{*,H} - z_i^*) +$$

$$\frac{1}{n} \sum_{i=1}^n (\underline{W}^* - \underline{W})^T \cdot (\hat{z}_i^{*,H} - z_i^{*,H}) =$$

$$\underbrace{\frac{1}{n} \sum_{i=1}^{n} (\underline{W}_i^* - \underline{W})^T \cdot (z_i^* - z_i)}_{(1)} + \underbrace{\frac{1}{n} \sum_{i=1}^{n} (\underline{W}^* - \underline{W}_i^*)^T \cdot (z_i^* - z_i)}_{(2)} +$$

$$\underbrace{\frac{1}{n} \sum_{i=1}^{n} (\underline{W}^* - \underline{W})^T \cdot (z_i^{*,H} - z_i^*)}_{(3)} + \underbrace{\frac{1}{n} \sum_{i=1}^{n} (\underline{W}^* - \underline{W})^T \cdot (\hat{z}_i^{*,H} - z_i^{*,H})}_{(4)}$$

**(1)**: Since the sub-optimality criterion implies a difference in value of at least $\epsilon$ for the initial distribution which assigns 1 to the state where the agent errs, we may use identical arguments to the previous proof. Therefore, the term is bounded from below by $\epsilon$.

**(2)**: By assumption $||\underline{W}^* - \underline{W}_i^*||_\infty \le \frac{(1-\gamma)\epsilon}{8k}$ thus since $||(z_i^* - z_i)||_1 \le \frac{2k}{1-\gamma}$ by Hölder's inequality the term is bounded by $\frac{\epsilon}{4}$.

**(3)**: We have $||x_i^{*,H} - x_i^*||_1 \le \frac{k\gamma^H}{1-\gamma}$ from definitions, thus $||z_i^{*,H} - z_i^*||_1 \le \frac{k\gamma^H}{1-\gamma}$ since $c \in \Delta_{d-1}$. As mentioned previously we may assume $||W^* - W_t||_\infty \le 2$, therefore by Hölder's inequality the term is bounded by $\frac{\epsilon}{4}$ due to our choice of $H$: $\gamma^H = (1 - (1-\gamma))^H \le \left((1 - (1-\gamma))^{\frac{1}{1-\gamma}}\right)^{(1-\gamma)H} = \left((1 - (1-\gamma))^{\frac{1}{1-\gamma}}\right)^{\log(\frac{8k}{(1-\gamma)\epsilon})} \le e^{-\log(\frac{8k}{(1-\gamma)\epsilon})} = \frac{(1-\gamma)\epsilon}{8k}$.

**(4)**: The partial sums $\sum_{i=1}^{N} (\underline{W}^* - \underline{W})^T \cdot (z_i^{*,H} - \hat{z}_i^{*,H})$ for $N = 0, ..., n$ form a martingale sequence. Note that $||z_i^{*,H}||_1 \le \frac{k}{1-\gamma}$, and $||\hat{z}_i^{*,H}||_1 \le \frac{k}{1-\gamma}$. Also, we have that $||W^* - W_t||_\infty \le 2$, thus, we can apply Azuma's inequality (Lemma 4) with $b = \frac{4k}{(1-\gamma)}$ and with our chosen n this yields: $\sum_{i=1}^{n} (\underline{W}^* - \underline{W})^T \cdot (z_i^{*,H} - \hat{z}_i^{*,H}) \le \frac{n\epsilon}{4}$ with probability of at least $1 - \frac{\delta}{2dk(dk+1)\log(\frac{16k\sqrt{dk}}{(1-\gamma)\epsilon})}$.

Thus $(\underline{W}^* - \underline{W})^T \cdot (\frac{\bar{Z}^*}{n} - \frac{\bar{Z}}{n}) > \frac{\epsilon}{4}$ and as in Theorem 3 this shows $B_\infty(\underline{W}^*, \frac{(1-\gamma)\epsilon}{8k})$ is never disqualified, and the number of updates is bounded by $2dk(dk+1)\log(\frac{16k\sqrt{dk}}{(1-\gamma)\epsilon})$, and multiplied by n this yields the upper bound on the number of rounds in which a sub-optimal action is chosen. By union-bound, the required bound for term **(4)** holds in all updates with probability of at least $1 - \delta$. $\qquad\square$

# E    CONTEXTUAL POLICIES

Consider the following problem. Given a complete specification of an MDP, and a hypothesis class $H$, for each state $s$ assign a hypothesis $h_a : C \to A$ such that the return is maximized.

The following theorem shows that it is NP-complete to find such a policy. We will use the class of linear separators. We define the following *contextual MDP training problem*. We are given an MDP with only transitions and a sample of $m$ contexts and their reward (namely for each context $c_i$ we specify the rewards for each state and action).

**Theorem 5.** *There is a reduction from training problem of the union of $k$ hyperplanes to the contextual MDP training problem of $k + 1$ states.*

*Proof.* Consider the following MDP, which has two parameters $r_0$ and $r_1$ which will define the rewards for each context. The MDP composed from a line of $k$ internal states $i$, and one sink state. Each internal state $i \in [1, k-1]$ has two actions: action 1 leads to the sink state with reward $r_1$ and the action 0 lead to the next internal state $i + 1$ with reward 0. In internal state $k$ action 1 leads to the sink state with reward $r_1$ and action 0 leads to the sink state with reward $r_0$. In the sink state there is a singe action that stays in the sink state and has reward 0.

The context is $C = \mathbb{R}^d$. The hypotheses class includes all hyperplanes, each hyperplane is characterize by a weight vector $w \in \mathbb{R}^d$, and if $w^\top c \geq 0$ then the action is 1 and otherwise it is 0.

Given a sample of size $m$ to the training problem $(c_i, y_i)$, we generate rewards for the contextual MDP training problem by having the context $c_i$ and specifying the parameters $r_0$ and $r_1$. Specifically, we will set $r_0 = 1 - y_i$ and $r_1 = y_i$.

Given $k$ hyperplanes $w_1, \ldots, w_k$, and $m$ example, assume that they make $e$ errors. By using those $k$ hyperplanes in the $k$ internal nodes for each context $c_i$ we have a reward 1 iff the union of the $k$ hyperplanes classify it correctly. This implies that we have reward $m - e$. This is since each example that is labeled correctly will get a reward of 1 and each incorrect example will get a reward of 0.

Given an assignment of $k$ hyperplanes to the internal nodes, which gets a return of $m - e$, we output as a hypothesis the union of the $k$ hyperplanes. Again, the number of errors on the sample is exactly $e$ errors. $\qquad\square$

