# OpenReview forum: "Contextual Inverse Reinforcement Learning"
_ICLR.cc/2020/Conference — Reject_

### Official Review · AnonReviewer3 · 2019-10-22
**Official Blind Review #3**

**Rating:** 3

**Review:**

This work focuses on the problem of 'contextual' inverse reinforcement learning, where the reward is a function of the current state of the MDP, and a set of context features, which remain constant within each episode.  The primary contribution of this work is the formulation of inverse reinforcement learning (for restricted spaces of context-dependent reward functions) as a convex optimization problem.  Based on this formulation, the paper describes several IRL algorithms based on approaches to solving convex and non-convex optimization problems, including variations of mirror descent, and evolution strategies (in principle allowing for the optimization of reward functions with arbitrary parametric representations).  The algorithms presented in this work all assume that computing an optimal policy for a specific reward function is a relatively inexpensive subroutine, which limits their applicability to domains where such planning is straightforward.  Experimental results are presented for a simple highway driving domain, as well as a simulated patient treatment domain constructed from real-world clinical data.

While the paper focuses one contextual IRL as its key contribution, the paper fails to sufficiently motivate the contextual IRL problem as a useful specialization of inverse reinforcement learning.  While there are clearly problems where a distinct 'context' can be identified, it isn't clear what advantage is gained by considering the context as a separate set of features, as opposed to treating the context as an additional set of state features which do not change within an episode.  With the latter formulation, many existing approaches to inverse reinforcement learning or imitation learning would be applicable.  The specific formulation of the reward as the product of the context and state features is somewhat more flexible than limiting the reward to linear functions of both, but it would be straightforward to apply the proposed algorithms to the case where the state and context are concatenated, and the reward is a quadratic function of state-context vector.

The experimental results are not particularly useful in evaluating the proposed algorithms, as the tasks involved are relatively simple, discrete-state MDPs (with continuous state features), and more importantly, no comparisons with existing IRL approaches are provided.  The potential scalability of evolution strategy optimization to more complex, non-convex reward representations such as deep networks is mentioned, but never empirically demonstrated.

**Experience Assessment:**

I have published one or two papers in this area.

**Review Assessment: Checking Correctness Of Derivations And Theory:**

I assessed the sensibility of the derivations and theory.

**Review Assessment: Checking Correctness Of Experiments:**

I carefully checked the experiments.

**Review Assessment: Thoroughness In Paper Reading:**

I read the paper at least twice and used my best judgement in assessing the paper.

---

> ### Author Response · Authors · 2019-11-09
> **Respond to Review #3**
>
> We would like to thank the reviewer for his time and effort in writing his review. The reviewer had concerns regarding the scalability of the COIRL algorithms and the lack of comparison with existing IRL methods. Let us address those issues.
>
> Scalability.
>
> By formulating the COIRL problem as a convex optimization problem we were able to derive subgradient based algorithms. We analyzed those algorithms in the linear case (for which we developed theoretical guarantees), but we would like to refer the reviewer to the experiments that we performed in the nonlinear setup using DNNs (figure 2). In these experiments, the subgradient of the loss is computed (instead of a gradient) and the rest of the DNN is differentiated with backpropagation. In addition, following the reviewer's response, we performed an additional experiment with a nonlinear mapping (but now in the offline medical regime), which can be found on the new version of the paper  (figure 4, and a short paragraph below it). We would like to point out that the theoretical derivation of the subgradient allowed us to make the extension to DNNs in a straight forward manner. We view the "move" from a theoretical understanding of the COIRL problem to a DNN implementation to be of high novelty and importance.
>
> As the COIRL problem is new,  we believe that formulating and providing theoretical guarantees to it is highly important. Considering the discrete state space MDP was crucial for comparing the sample complexity results with those of previous IRL/AL papers. Finally, from a theoretical point of view, introducing the exponential weights algorithm allowed us to obtain algorithms that scale well to problems with high dimensional feature and context spaces and also makes the COIRL problem to be computationally comparable with standard IRL.
>
> Comparison with existing IRL methods.
> We have added a section in which we compare COIRL to IRL in a large MDP (section 3.4). In particular, we discuss how the contextual model significantly reduces the cost of the inner loop (in which an optimal policy is computed). This is what makes the COIRL feasible in high-dimensional problems where standard IRL algorithms may fail due to this exact issue.
> In the new section mentioned previously, we provide a comparison with standard IRL in order to highlight the advantages of COIRL. We will also add simulations for this large MDP model shortly.

---

> > ### Comment · AnonReviewer3 · 2019-11-14
> > **Novelty of COIRL**
> >
> > Thank you for taking the time to address these concerns.
> >
> > Ultimately the COIRL problem is not particularly novel in the imitation learning space.  Several authors have considered imitation of policies with latent contexts:
> >
> > Li, Yunzhu, Jiaming Song, and Stefano Ermon. "Infogail: Interpretable imitation learning from visual demonstrations." Advances in Neural Information Processing Systems. 2017.
> >
> > Babes, Monica, et al. "Apprenticeship learning about multiple intentions." Proceedings of the 28th International Conference on Machine Learning (ICML-11). 2011.
> >
> > As well as the case with observable contexts (such as natural language instructions):
> >
> > Misra, Dipendra, John Langford, and Yoav Artzi. "Mapping instructions and visual observations to actions with reinforcement learning." arXiv preprint arXiv:1704.08795 (2017).
> >
> > It is important to compare your methods experimentally against existing baselines for non-contextual imitation learning.  At the very least, comparisons could be made against behavioral cloning, where the policy is conditioned on both the state and the context.
> >
> > Also, I don't believe that section 3.4 is visible in the current version.  How does the COIRL formulation reduce the cost of computing the optimal policy?

---

> > > ### Author Response · Authors · 2019-11-14
> > > **Quick response to AnonReviewer3**
> > >
> > > We thank the reviewer for his follow up. We will address the reviewers comments soon. On the meanwhile, we would like to refer the reviewer to section 3.4 as he requested.
> > >
> > > The up-to-date version of our paper includes:
> > > Section 5 (previously 3.4): comparison with related work - theory and experiments.
> > > DL experiments in the medical offline domain: figure 4.
> > >
> > > - the authors

---

> > > > ### Author Response · Authors · 2019-11-14
> > > > **Follow up**
> > > >
> > > > Following up on our last comment, we would like to further address the reviewer's concerns regarding the novelty of our work.
> > > >
> > > > Indeed, various authors proposed to learn contextual policies in the RL and IRL regimes. In our paper, we mentioned two examples, Xu et al. (2018), and Fu et al. (2019). We thank the reviewer for providing additional references, and we will be happy to add them in a later revision. At the beginning of Section 5, we have added a theoretical explanation for why learning a contextual policy is an (NP-complete) hard problem from a computational perspective. While this explanation should not deter practitioners from trying to learn such policies (there probably exist many cases where this should be possible), from a theoretical point of view, there also exists cases where learning such a policy should not work. To the best of our knowledge, we provided the first algorithm with theoretical guarantees and final sample complexity in this setting.
> > > >
> > > > More specifically, the first two papers that the reviewer mentioned (Yunzhu et al. and Babes et al.) do not follow the COIRL problem formulation. The context of this problem is assumed to be hidden or latent. The approach that is taken follows a "hierarchical" formulation: at first, a cluster (contexts) is identified, and then a cluster-specific policy is learned for each context. By design, these methods assume a finite context space while our model works in continuous context spaces. The third work by Dipendra et al. learns a policy that is a function of both the context and the state, which we already argued to be NP-hard.
> > > >
> > > > Our approach avoids these difficulties by taking a model-based approach -- learning a reward function that is a function of the context and inferring the reward for each context.
> > > > The downside of our approach is that we need to compute the optimal policy for each context. However, as we have argued, we focus on the regime where the context space is rich and diverse, and the MDP is relatively small (please see a detailed comment about it in our response to reviewer 4). Thus, from a computational point of view, our approach is highly efficient in this setup (see Section 5 for discussion and examples).
> > > >
> > > > Kelvin Xu, Ellis Ratner, Anca Dragan, Sergey Levine, and Chelsea Finn. Learning a prior over intent via meta-inverse reinforcement learning, arXiv preprint arXiv:1805.12573, 2018.
> > > >
> > > > Justin Fu, Anoop Korattikara, Sergey Levine, and Sergio Guadarrama. From language to goals: Inverse reinforcement learning for vision-based instruction following. arXiv preprint arXiv:1902.07742, 2019.

---

> > > > > ### Comment · AnonReviewer3 · 2019-11-15
> > > > > **Hardness Results**
> > > > >
> > > > > The NP-Hardness results in section 5 either need to be expanded upon, or referenced.  For a known MDP (known dynamics and rewards) with finite states and actions, an optimal policy can be found via linear programming in polynomial time (Puterman., M. L. (1994) I believe), but it isn't clear whether you are referring to a known MDP or one where the transition probabilities must be inferred (the RL setting), or an MDP with finite states and continuous context vectors.
> > > > >
> > > > > From a practical perspective, the most straightforward baseline approach to these imitation learning problems would be behavioral cloning, learning a mapping directly from states and contexts to actions via supervised learning on the demonstrated state-action pairs, without attempting to infer the latent reward function the expert is optimizing.  Behavioral Cloning would seem to be applicable to all of he experimental domains considered here, and given enough demonstrations may perform similarly to the algorithms presented in this work much less computational complexity.

---

> > > > > > ### Author Response · Authors · 2019-11-15
> > > > > > **Hardness results**
> > > > > >
> > > > > > Please refer to appendix E in the uploaded version for an official proof of the NP hardness results. We believe that the proof also applies to the case of behavioural cloning.
> > > > > >
> > > > > > These results refer to the case that the policy is a set of linear classifiers (one in each state) that are all a function of the context. So for a given context, each one of these classifiers has to output the correct action to compose the optimal policy.
> > > > > >
> > > > > > We agree with the reviewer that from a practical perspective, learning a contextual policy is plausible and might work well in specific applications. However, in this paper we proposed a "from theory to practice" approach, i.e., a method that has both theoretical guarantees and can be extended to practical applications (by introducing DNNs).
> > > > > >
> > > > > > We also promise to add simulations comparing our method with behavioural cloning in a later version of this paper.
> > > > > >
> > > > > > -the authors

---

> > > > > > > ### Comment · AnonReviewer3 · 2019-11-15
> > > > > > > **Response**
> > > > > > >
> > > > > > > So the proof should make it clear exactly which NP-Complete problem you are reducing to the contextual MDP, describe the problem being reduced in detail, and provide a reference for this problem where it is shown to be NP-Complete.  Right now the proof is not detailed enough to follow, and it seems that solving the contextual chain-MDP would be strightforward.
> > > > > > >
> > > > > > > For behavioral cloning, the computational complexity of selecting the optimal hypothesis will of course depend on the exact hypothesis space being used.  For binary actions however, selecting a hyperplane which matches observed actions in a given state across several contexts would essentially be a problem of logistic regression.

---

> > > > > > > > ### Author Response · Authors · 2019-11-15
> > > > > > > > **Response**
> > > > > > > >
> > > > > > > > Thank you for following up on this.
> > > > > > > >
> > > > > > > > We promise to polish the proof in a future revision. However, due to the (soon) upcoming deadline of the rebuttal stage, we decided to upload it as it is to initiate discussion.
> > > > > > > >
> > > > > > > > For the behavioural cloning example, the reviewer is correct - the problem is equivalent to putting a logistic regression classifier in EACH STATE. The NP-hardness result stems from the fact that ALL of the classifiers need to classify correctly for EACH context. Thus, the reduction is to the problem of the union of k hyperplanes. This problem is known to be NP-hard and we promise to add the references in a future revision.

---

### Official Review · AnonReviewer1 · 2019-10-24
**Official Blind Review #1**

**Rating:** 6

**Review:**

The authors consider the problem of inverse reinforcement learning for CMDPs in which the reward function is a function of the context. They  propose different algorithms for learning the function and evaluate their algorithms on a driving simulator and a sepsis treatment problem (based on real data from the MIMIC corpus).

I think this is a good paper, studying an interesting problem and proposing useful solutions, so it should be accepted. The paper could be improved by putting more focus on the presentation of the studied problem and its variants (also the ones the authors mention their approach is easy to generalize to) and reducing a little the focus on the algorithm. In total they propose three algorithms and give convergence guarantees for all of them. Of course this analysis is important but I found it somewhat distracting from the main flow of the paper. Maybe these guarantees could be moved into a separate section.

A few more points:
* Please comment on why you see estimating contextual transition dynamics as an orthogonal problem.
* In the last paragraph of the conclusion you talk about a safety critical application. For which applications do you think this is practical? I would assume that constantly reviewing of an AI systems' actions is very impractical. (But I agree on these aspects being important.)
* One of the main limitations of this work seems to be that the CMDP\M has to be known. Please comment on how one could expand the analysis/applications/experiments to extend to the case where the CMDP is not known.


**Experience Assessment:**

I have published one or two papers in this area.

**Review Assessment: Checking Correctness Of Derivations And Theory:**

I assessed the sensibility of the derivations and theory.

**Review Assessment: Checking Correctness Of Experiments:**

I assessed the sensibility of the experiments.

**Review Assessment: Thoroughness In Paper Reading:**

I made a quick assessment of this paper.

---

> ### Author Response · Authors · 2019-11-09
> **Response to Review#1**
>
> Thank you for the detailed and insightful review.
>
> In the case of RL in CMDPs, an efficient algorithm for estimating the dynamics was proposed in [1] (and in other papers we mentioned in the paper). In the case of IRL, the dynamics are usually learned from samples before an IRL/AL algorithm is used (as in MWAL/Abeel&Ng). In COIRL, we can use the same method in [1] to learn the contextual dynamics before to the inverse learning step, in which we use the dynamics to calculate feature expectations for all contexts (which is the only place where the dynamics are needed). Our algorithms are otherwise unaffected by contextual dynamics. For these reasons, we consider estimating the dynamics to be an orthogonal/precursory problem, but we will happy to change/make this comment more clear.
>
> Safety-critical applications.
> While constantly reviewing AI systems' actions is impractical, we believe that in such safety-critical applications it is actually essential in the foreseeable future. In the medical treatment regime using ML without doctor supervision is not likely to happen soon. Compared to what is currently being introduced in the industry - doctors using ML-powered tools, e.g. detection of cancerous growths in images - our method does shift the load and responsibility slightly towards the agent.
> We would also like to comment that the subgradient based algorithms do not require such supervision so essentially, if this supervision is not a constraint, they should be preferred,
>
> Finally, we believe that our methods can be extended to problems that do not necessarily have a tabular state space and where the dynamics are unknown. This can be achieved, for example, by using DRL algorithm to compute the optimal policy for each context while a different network is learning the mapping from contexts to rewards. To improve the computational complexity of the algorithm, we believe that introducing  MAML [2] to find good initialization to the policy network will be crucial to success. This is an exciting research direction for future work and we are actively working on it.
>
> [1] Aditya Modi, Nan Jiang, Satinder Singh, and Ambuj Tewari. Markov decision processes with continuous side information. In Algorithmic Learning Theory, pp. 597–618, 2018.
> [2] Chelsea Finn, Pieter Abbeel, and Sergey Levine. Model-Agnostic Meta-Learning for Fast Adaptation of Deep Networks. In International Conference of Machine Learning 2017.

---

### Official Review · AnonReviewer4 · 2019-10-30
**Official Blind Review #4**

**Rating:** 6

**Review:**

This paper introduces a formulation for the contextual inverse reinforcement learning (COIRL) problem and proposed three algorithms for solving the proposed problem. Theoretical analysis of scalability and sample complexity are conducted for cases where both the feature function and the context-to-reward mapping function are linear. Experiments were conducted in both a simulated driving domain and a medical treatment domain to compare the three proposed algorithms empirically. Empirical results for using a deep network as the contextual mapping function is also provided.

As a special case of IRL for POMDPs, the contextual IRL problem (with latent contexts) is an interesting research topic that is of interest to researchers working on generalizing IRL to a greater range of real-world applications. This paper was written with clarity and detailed descriptions of experiments. The authors presented their algorithms with thorough theoretical analysis. However, further ablation study and proofs are needed to demonstrate the proposed problem formulation and algorithms outperform existing IRL frameworks.

The authors motivate the problem of COIRL from the formulation of contextual MDP by Hallak et al. (2015). However, the referenced work of Hallak et al. (2015) (published as a preprint on arXiv) does not provide a sufficient argument/evidence for why or in what cases this particular formulation, especially when the context is observed (as assumed by this paper), is better than alternative ones, such as directly modeling the context as part of the state. When a discrete number of contexts exist (as in the simulated driving domain: average vehicle v.s. ambulance), the proposed problem can be reduced to solving a set of normal IRL problems, i.e. learning a reward function under each observed context. For continuous context variables (such as age and weight for patients in the medical treatment domain), an intuitive solution is to model them as part of the state/feature and run normal IRL algorithms. However, the authors did not analyze or show in either of their experiments how existing IRL algorithms compare with the proposed algorithms. Without further explanation, the authors claim, in the fourth paragraph of section 1, that “..Apprenticeship Learning algorithms that try to mimic the expert cannot be used..”, yet it is not trivial to understand why the proposed formulation and algorithms would outperform existing methods/baselines.

Lastly, it would be more insightful to the readers if the authors can provide some discussion on how they would extend their algorithms to the more interesting/practical case where the context is not directly observed and analyze how latent contexts would affect the performance/complexity of their proposed algorithms.

Overall, this is a well-written paper on an exciting research topic but lacks sufficient analysis and experimental results to support the significance of the intended contributions.


Reference (from the paper):

Hallak, A., Di Castro, D., & Mannor, S. (2015). Contextual Markov decision processes. arXiv preprint arXiv:1502.02259.


==================================


Edit after rebuttal period:

- The updated version of the paper includes preliminary results of comparisons with alternative methods and shows that COIRL does have better scalability for modeling problems with large number of observable contexts. It would be great if the authors can show how the baseline performs in the two original experiments as well to understand the improvements gained from adopting COIRL for these problems.

- As stated before, I think COIRL is an exciting research topic on its own. While many recent works on IRL focuses on generalizing to complex problem with deep neural nets as function approximation tools, it is important to analyze the underlying problems from bottom up (in smaller domains and with assumptions that may not always hold) in order to better understand the limits of different solutions. I'd certainly love to see more follow-up works on this topic and how it can be extended to more complex problems.

- Hence, after carefully reading the authors responses and the updated version of the paper, I changed my decision to 'Weak Accept'.



**Experience Assessment:**

I have published one or two papers in this area.

**Review Assessment: Checking Correctness Of Derivations And Theory:**

I assessed the sensibility of the derivations and theory.

**Review Assessment: Checking Correctness Of Experiments:**

I carefully checked the experiments.

**Review Assessment: Thoroughness In Paper Reading:**

I read the paper thoroughly.

---

> ### Author Response · Authors · 2019-11-09
> **Response to Review#4**
>
> We would like to thank the reviewer for his time and effort in writing his review.
>
> Comparison with existing IRL methods.
> We have added a section in which we compare COIRL to IRL in a large MDP (section 3.4). In particular, we discuss how the contextual model significantly reduces the cost of the inner loop (in which an optimal policy is computed). This is what makes the COIRL feasible in high-dimensional problems where standard IRL algorithms may fail due to this exact issue.
> In the new section mentioned previously, we provide a comparison with standard IRL in order to highlight the advantages of COIRL. We will also add simulations for this large MDP model shortly.
>
> We would also like to note that after they were initially proposed in the mentioned arXiv paper,
> CMDPs have been studied in [1,2,3,4] who provided theoretical guarantees for RL algorithms in this setup (and were published in top ML venues).
>
> Regarding continuous state/action spaces, we focused on the case that the context is observed. We believe that this is a case of great practical interest, for instance, the road conditions and weather which affects the vehicle’s trajectory. We would like to refer the reviewer to [6] who studied this problem and derived hardness (NP-complete) results.
>
> We believe that our methods can be extended to problems that do not necessarily have a tabular state space and where the dynamics are unknown. This can be achieved, for example, by using DRL algorithm to compute the optimal policy for each context while a different network is learning the mapping from contexts to rewards. To improve the computational complexity of the algorithm, we believe that introducing  MAML [5] to find good initialization to the policy network will be crucial to success. This is an exciting research direction for future work and we are actively working on it.
>
> [1] Abbasi-Yadkori, Yasin, and Gergely Neu. "Online learning in MDPs with side information." arXiv preprint arXiv:1406.6812 (2014).
> [2] Modi, Aditya, et al. "Markov decision processes with continuous side information." Algorithmic Learning Theory. 2018.
> [3] Dann, Christoph, et al. "Policy certificates: Towards accountable reinforcement learning." In International Conference of Machine Learning 2019.
> [4] Modi, Aditya, and Ambuj Tewari. "Contextual markov decision processes using generalized linear models." ICML 2019 Workshop on RL for Real Life.
> [5] Chelsea Finn, Pieter Abbeel, and Sergey Levine. Model-Agnostic Meta-Learning for Fast Adaptation of Deep Networks. In International Conference of Machine Learning 2017.
> [6] Steimle, Lauren N., David L. Kaufman, and Brian T. Denton. "Multi-model Markov decision processes." Optimization Online URL http://www. optimization-online. org/DB_FILE/2018/01/6434. pdf (2018).

---

> > ### Comment · AnonReviewer4 · 2019-11-14
> > **Experiments**
> >
> > Thanks the authors for their time addressing the concerns.
> >
> > The new results show that, under the given CMDP formulation for 3x4 gridworlds, the proposed COIRL algorithm outperforms the projection algorithm in terms of scalability to increasing number of contexts. However, there is no result showing how the baseline would compare in the two original experiments, which do not have too many contexts. Please comment on what class of practical problems, that the authors think COIRL is applicable to, would benefit the most from the COIRL formulation.
> >
> >
> > Thanks

---

> > > ### Author Response · Authors · 2019-11-14
> > > **Class of practical problems**
> > >
> > > We want to thank the reviewer for following up on our response.
> > >
> > > The COIRL formulation fits the most to problems that have a large context space and a relatively "small MDP".  While our work is motivated by applications in the medical regime, we believe that the model also fits various web applications such as personalized advertisement, conversational chatbots, etc. All of these problems share the following properties: 1) Planning is essential - there is more than a single decision to make, and the consequences of decisions impact later decisions (in contrast to supervised learning). 2) Planning for a specific user is not that hard -- once the context is known, each MDP is solvable. 3) The user/context space is rich and diverse: there are many different users, and each one of them requires a specialized treatment.
> > >
> > > Specifically, in the dynamic treatment regime, there are many contexts. In our experiments, there are thousands of demonstrations, and each one of these demonstrations corresponds to a unique and different patient (context).
> > >
> > > We would also like to mention that this paper takes a bottom-up approach. We believe that understanding the fundamental mechanics of COIRL in the ״tabular MDP - large context regime״, theoretically and empirically, will allow us to move forward later to the large MDP regime. In the paper, we demonstrated that our method could be extended to the setting that the mapping from contexts to rewards is nonlinear, and we hope to extend our results to more regimes in future work. Nevertheless, we believe that the ״tabular MDP - large context regime״  is attractive on its own and has exciting use cases, as we mentioned above.
> > >
> > > the authors

---

### Author Response · Authors · 2019-11-12
**New revision**

We have uploaded a revised version of the paper that addresses some of the comments raised by the reviewers.

This version includes:
Section 5 (previously 3.4): comparison with related work - theory and experiments.
DL experiments in the medical offline domain: figure 4.

- the authors

---

### Decision · Program_Chairs · 2019-12-19

**Decision:**

Reject

**Comment:**

The authors introduce a framework for inverse reinforcement learning tasks whose reward functions are dependent on context variables and provide a solution by formulating it as a convex optimization problem.  Overall, the authors agreed that the method appears to be sound.  However, after discussion there were lingering concerns about (1) in what situations this framework is useful or advantageous, (2) how it compares to existing, modern IRL algorithms that take context into account, and (3) if the theoretical and experimental results were truly useful in evaluating the algorithm.  Given that these issues were not able to be fully resolved, I recommend that this paper be rejected at this time.